# Random Spiking Neural Networks are Stable and Spectrally Simple

**Ernesto Araya** [*]
Department of Mathematics,
Ludwig-Maximilians-Universität München
Munich Center for Machine Learning (MCML)
araya@math.lmu.de

**Massimiliano Datres** [*]
Department of Mathematics,
Ludwig-Maximilians-Universität München
Munich Center for Machine Learning (MCML)
datres@math.lmu.de

**Gitta Kutyniok**
Department of Mathematics,
Ludwig-Maximilians-Universität München
Munich Center for Machine Learning (MCML)
University of Tromsø
DLR-German Aerospace Center
kutyniok@math.lmu.de

## Abstract

Spiking neural networks (SNNs) are a promising paradigm for energy-efficient computation, yet their theoretical foundations—especially regarding stability and robustness—remain limited compared to artificial neural networks. In this work, we study discrete-time leaky integrate-and-fire (LIF) SNNs through the lens of Boolean function analysis. We focus on noise sensitivity and stability in classification tasks, quantifying how input perturbations affect outputs. Our main result shows that wide LIF-SNN classifiers are stable on average, a property explained by the concentration of their Fourier spectrum on low-frequency components. Motivated by this, we introduce the notion of *spectral simplicity*, which formalizes simplicity in terms of Fourier spectrum concentration and connects our analysis to the *simplicity bias* observed in deep networks. Within this framework, we show that random LIF-SNNs are biased toward simple functions. Experiments on trained networks confirm that these stability properties persist in practice. Together, these results provide new insights into the stability and robustness properties of SNNs.

## 1 Introduction

Artificial Neural Networks (ANNs) have become central to modern machine learning, but their rapid growth in scale demands increasingly unsustainable computational and energy resources (Thompson et al., 2021). This challenge is especially acute for low-power devices, where efficiency is critical. Spiking Neural Networks (SNNs), inspired by biological neurons and operating through event-driven spikes, offer a promising energy-efficient alternative (Roy et al., 2019; Davies et al., 2018). Their sparse communication and compatibility with neuromorphic hardware position them as strong candidates for sustainable machine learning (Mehonic et al., 2024; Fono et al., 2025).

Despite this potential, the theoretical understanding of SNNs remains limited compared to that of classical ANNs. While substantial progress has been made on designing training algorithms (Eshraghian et al., 2023) and hardware implementations (Davies et al., 2018; Indiveri & Liu, 2015), core theoretical properties—such as stability, robustness, and generalization—are still largely underexplored. This gap limits our ability to rigorously assess the strengths and limitations of SNNs in practice.

---

[*]Equal contribution.

Among these properties, stability is crucial for designing networks resilient to input perturbations and adversarial attacks, yet no single definition exists. It can refer to algorithmic stability of learning algorithms (Elisseeff et al., 2005), dynamical systems stability (naturally aligned with the temporal dynamics of spiking neurons) (Ding et al., 2024), or—as we consider here—to sensitivity to input changes. Intuitively, a stable network should be resilient to small perturbations in inputs or parameters, which is critical for both reliable inference and efficient learning. Many SNN models, including the discrete-time Leaky Integrate-and-Fire (LIF) neuron, can be viewed as iterative compositions of Boolean functions, since neurons emit spikes only when their membrane potential crosses a threshold (a binary event), motivating the study of stability via Boolean function analysis (O'Donnell, 2014). While prior work has examined SNN stability from dynamical systems (Ding et al., 2024) or neuroscience perspectives (Calaim et al., 2022), to our knowledge this is the first study to apply Boolean analysis to characterize and investigate stability in SNNs.

The Boolean perspective on SNNs suggests a natural link to the notion of simplicity in neural networks: if a network is stable, small input perturbations rarely change its output, hinting at a bias toward "simple" input-output mappings. In the SNN setting, this can be formalized via the predominance of low-frequency components in the Fourier–Walsh expansion of a classifier. Motivated by this, we introduce *spectral simplicity*, which quantifies the concentration of the Fourier spectrum on low frequencies. This notion connects naturally with other notions of simplicity studied in the context of *simplicity bias*, proposed as a lens to understand generalization in deep networks. For example, prior work has shown that random deep ANNs tend to implement "simple" functions (De Palma et al., 2019), formalized as a large average Hamming distance to the nearest input with a different predicted class (Valle-Pérez et al., 2019). Spectral simplicity represents a weaker notion, but one that arises intrinsically in spiking networks.

To make these questions precise, we focus on the discrete-time LIF model. This model combines theoretical simplicity with widespread adoption in practice, serving as the foundation of software frameworks such as SNN-Torch and being implemented in digital neuromorphic platforms including Loihi 2 and SpiNNaker 2 (Orchard et al., 2021; Gonzalez et al., 2023). In this work, we restrict attention to networks at initialization. This choice reflects the still-developing theory of SNN training and our goal of isolating stability properties intrinsic to the model, without confounding effects from learning dynamics. Moreover, random networks have been shown to serve as useful priors in PAC-Bayes generalization bounds (Valle-Pérez et al., 2019). Understanding the effect of training dynamics is an exciting question; here, we explore it experimentally and leave a theoretical treatment for future work.

**Contributions.** We summarize our main contributions as follows:

- We derive quantitative bounds on the stability of discrete-time LIF SNN classifiers, showing that they are stable on average with respect to random parameter initialization. In particular, when input sequences lie in the binary cube $\{-1, 1\}^n$, the classifier output remains unchanged under perturbations of up to $\mathcal{O}(\sqrt{n})$ coordinates, with high probability for $n$ large enough. The bounds depend explicitly on the model's hyperparameters and reveal that LIF SNNs exhibit stability properties comparable to other (time-independent) Boolean networks (Jonasson et al., 2023).

- We introduce the notion of *spectral simplicity*, defined via the Fourier decomposition of discrete-time LIF SNN classifiers for static data. We prove that random SNNs are biased toward spectrally simple functions, i.e., those whose Fourier spectrum is predominantly concentrated on low-frequency components.

- We complement these theoretical results with numerical experiments, investigating in particular the effect of training on stability. Our experiments reveal that both shallow and deep SNNs are noise stable, and that training tends to increase their stability on average.

## 1.1 RELATED WORKS

**Stability of ANNs and SNNs.** As previously noted, stability in SNNs can be approached from multiple perspectives. Each neuron follows a (typically non-linear) dynamical system, raising the natural question of how input perturbations affect its output. In the ANN and Neural ODE literature, Lyapunov-based analyses (Jimenez-Rodriguez et al., 2022; Kang et al., 2021; Rahnama et al.,

2019) establish robustness via bounds on how perturbations propagate through the network. The discrete-time LIF model's robustness to input perturbations has been analyzed from a dynamical systems perspective by Ding et al. (2024), who derived bounds on output spike sequence variations under perturbed inputs. Our work differs in three key ways: (i) we study stability at the classifier level, where predictions may remain unchanged even if spike trains differ; (ii) we consider the reset-by-subtraction mechanism, which introduces additional complexity compared to the reset-to-zero simplification in (Ding et al., 2024); and (iii) our analysis extends beyond single neurons to multi-neuron networks. Methodologically, our approach is different building on Boolean function analysis (O'Donnell, 2014), aligning more closely with Jonasson et al. (2023), but introduces new challenges stemming from reset dynamics and temporal evolution, which create nontrivial probabilistic dependencies.

**Simplicity bias.** Simplicity bias has been proposed as a mechanism underlying the generalization of deep networks, both in trained models with SGD (Arpit et al., 2017; Nakkiran et al., 2019; Valle-Pérez et al., 2019) and in random ANNs (De Palma et al., 2019). The central idea is that learning favors simple functions, though the precise definition of simplicity varies, and pitfalls have been noted in (Shah et al., 2020). We introduce *spectral simplicity*, based on the Fourier–Walsh decomposition of Boolean functions, as a complementary notion. While our focus on random networks parallels (De Palma et al., 2019), the techniques differ substantially: their analysis relies on Gaussian process arguments, whose extension to SNNs is unclear, whereas we draw on Boolean function analysis, which adapts naturally to our setting. While a weaker measure of simplicity, spectral simplicity arises organically in spiking networks.

## 1.2 NOTATION

Give a positive integer $a$, we denote with $[a]$ the sets $\{1, \ldots, a\}$. Given $x, y \in \{-1, 1\}^n$, we define the Hamming distance between $x$ and $y$ as $d_H(x, y) := \left|\left\{i \in [n] : x_i \neq y_i\right\}\right| = \frac{1}{2}\sum_{i=1}^{n}|x_i - y_i|$. We use $\text{sign}(x) \in \{-1, 1\}$ for the sign function, i.e. $\text{sign}(x) = 1$ if $x \geq 0$ and $-1$ otherwise. We write $\mathcal{N}(u, \Sigma)$ for the Gaussian distribution with mean $u$ and covariance $\Sigma$, $\text{Unif}(\{-1, 1\}^n)$ for the uniform distribution on the hypercube, and $\text{Bin}(n, p)$ for the Binomial distribution with parameters $(n, p)$. Similarly, $\text{Rad}(\kappa)$ denotes a vector with i.i.d. Rademacher coordinates with parameter $\kappa$(dimension clear from context). For asymptotics, we use the standard asymptotic notation: $\mathcal{O}(\cdot)$, $o(\cdot)$ and $\omega(\cdot)$.

## 2 THE DISCRETE-TIME LIF MODEL

We aim to study the stability of SNNs in classification tasks. Specifically, we consider SNN models constructed as compositions of *sign leaky integrate-and-fire* (sLIF) neurons. Each neuron acts as an information-processing unit that maps time series inputs[1] $(x_t)_{t\in[T]} \in (\{-1, 1\}^n)^T$ to binary spike sequences $(s_t)_{t\in[T]} \in \{-1, 1\}^T$, based on a neuronal dynamic. The dynamics of a single neuron are defined as follows.

**Definition 1** (sLIF neuron). *Let $T \geq 1$ be an integer, $\beta \in [0, 1]$, $\theta \in (0, \infty)$. We define the sign leaky integrate-and-fire (sLIF) neuron, with input $(x_t)_{t\in[T]} \in (\{-1, 1\}^n)^T$ and output $(s_t)_{t\in[T]} \in (\{-1, 1\})^T$, as a parametric computational unit that evolves over discrete time steps $t \in [T]$ accordingly to the following recursive dynamic*

$$\begin{cases} u_t = \beta u_{t-1} + w^\top x_t - \frac{\theta}{2}\left(s_{t-1} + 1\right) \\ s_t = \text{sign}\left(u_t - \theta\right) \\ u_0 = 0 \end{cases}, \tag{1}$$

*where $(u_t)_{t\in[T]} \in ([0, \infty))^T$ is the sequence of membrane potentials and $w \in \mathbb{R}^n$ are the model's weights. Equivalently, this can be regarded as a function mapping $(x_t)_{t\in[T]} \rightarrow (s_t)_{t\in[T]}$. To make the dependence on inputs and parameters explicit, we occasionally use the notation $s_t\left((x_k)_{k\in[t]}, w\right)$.*

---

[1]While our theory focuses on binary data, the definition extends to real-valued inputs.

Each neuron processes an input sequence $(x_t)_{t\in[T]}$ through its membrane potentials $(u_t)_{t\in[T]}$, which evolve according to an autoregressive decay dynamics over the time horizon $T$ (referred to as the *latency* of the model). The *leak parameter* $\beta \in [0,1]$ controls this decay, specifying the fraction of potential retained per time step. Whenever the membrane potential exceeds the activation threshold $\theta > 0$ at some time $t'$, the neuron emits a spike, recorded as $s_{t'} = 1$ in the spike train $(s_t)_{t\in[T]}$, and the potential is reduced by $\theta$ (*reset by subtraction*). Weights are initialized as $w \sim \mathcal{N}(0, I_n/n)$, ensuring $w^\top x = O(1)$ with high probability for $x \in \{-1,1\}^n$, thereby avoiding degenerate regimes of vanishing or overly frequent firing. For further background on the LIF model, see (Gerstner & Kistler, 2002). In short, we use the term *leaky integrate-and-fire (LIF) neuron* for the variant with a Heaviside step function and reset rule $v_t \mapsto v_t - \theta s_{t-1}$, and *integrate-and-fire (IF)* when the leakage is absent ($\beta = 1$); networks composed of such units are referred to as LIF and IF SNNs, respectively.

**Networks of spiking neurons.** In the sLIF SNNs considered here, multiple sLIF neurons are interconnected through weighted synapses. The spike train generated by each neuron can serve as input to other neurons, and the overall network dynamics evolve over time, akin to a recurrent network. In what follows, we focus on SNNs composed of fully connected sLIF neurons, arranged in layers, as formalized in the next definition.

**Definition 2** (sLIF SNN). *Fix positive integers $L, T, n_0, \ldots, n_L \geq 1$, $\beta \in [0,1]$, $\theta \in [0,\infty)$, and let the input sequence be $(x_t)_{t\in[T]} \in (\{-1,1\}^{n_0})^T$. An $L$-layer sLIF neural network with latency $T$ and layer widths $n_1, \ldots, n_L$ is defined as the Boolean dynamical system*

$$\begin{cases} u_t^{(l)} = \beta u_{t-1}^{(l)} + W^{(l)} s_t^{(j-1)} - \frac{\theta}{2}\big(s_{t-1}^{(l)} + 1\big), \\ s_t^{(l)} = \operatorname{sign}\big(u_t^{(l)} - \theta\big), \\ u_0^{(l)} = 0, \\ s_t^{(0)} = x_t, \end{cases} \tag{2}$$

*where, for each layer $l \in [L]$, $W^{(l)} \in \mathbb{R}^{n_l \times n_{l-1}}$ denotes the weight matrix, $(u_t^{(l)})_{t\in[T]}$ the membrane potential sequence, and $(s_t^{(l)})_{t\in[T]}$ is the (output) the spike sequence. We collect all parameters into a single vector $W = \operatorname{vec}\big(W^{(1)}, \ldots, W^{(L)}\big) \in \mathbb{R}^d$, where $d = \sum_{l=1}^L n_l n_{l-1}$.*

Given a $L$-layers sLIF SNN as in Definition 2, a widely used choice of classifier in SNNs (see e.g., (Diehl & Cook, 2015)) is based on spike counts at the output layer, namely

$$f^{L,T}\big((x_t)_{t\in[T]}, W\big) := \arg\max_{i\in[n_L]} \sum_{t=1}^T s_{t,i}^{(L)}\big((x_t)_{t\in[T]}, W\big), \tag{3}$$

where the predicted class corresponds to the neuron in the final layer with the largest total spike count. Here, $s_{t,i}^{(L)}$ is the $i$-th coordinate of the spiking sequence $s_t^{(L)}$.

**Assumptions.** Throughout this work, we assume $n_0 = \cdots = n_{L-1} = n$, with $n_L$ equal to the number of classes. The extension of the subsequent results in the case $\beta \neq 1$ is discussed in Appendix D. For simplicity, we focus on a variant of the sLIF model equation 2 with $\beta = 1$. Our analysis allows dynamic input sequences $(x_t)_{t\in[T]}$, though some results only apply to static inputs, interpreted as repeated presentations of the same sample over time. Such constant input encoding is commonly used in practice for time-static datasets such as MNIST or CIFAR-10 (Rathi & Roy, 2023; Rueckauer et al., 2017). Finally, the model parameters are initialized as

$$W_i \overset{i.i.d.}{\sim} \mathcal{N}(0, 1/n), \quad i \in [d].$$

## 3 NOISE SENSITIVITY OF BOOLEAN FUNCTIONS

Within this framework, sLIF neurons can be viewed as compositions of Boolean functions, allowing stability analysis via Boolean function theory. We recall the basic definitions here and refer the reader to (O'Donnell, 2014) for a comprehensive treatment. Additionally, we introduce *spectral simplicity*, which is central to the statement of our main result.

**Noise sensitivity and stability.** The stability of a Boolean function is classically quantified by its *noise sensitivity*. For $f : \{-1,1\}^n \to \{-1,1\}$ and noise rate $\nu \in [0,1]$, define

$$\mathbf{NS}_\nu(f) := \mathbb{P}_{x,\xi}[f(x) \neq f(x \odot \xi)],$$

where $x \sim \mathrm{Unif}(\{-1,1\}^n)$ and $\xi = (\xi_1, \ldots, \xi_n)$ has i.i.d. entries $\xi_i \sim \mathrm{Rad}(1-\nu)$. Equivalently, the *noise stability* is

$$\mathbf{Stab}_{1-2\nu}(f) := \mathbb{E}_{x,\xi}[f(x)f(x \odot \xi)] = 1 - 2\,\mathbf{NS}_\nu(f),$$

capturing the probability that $f$ preserves its value under input perturbations. Given this equivalence, we will use noise sensitivity in the sequel.

**Definition 3** (Expected noise sensitivity). *For a parametric family $\{f_w\}_{w \in \mathcal{W}}$, a probability measure $\mu$ on $\mathcal{W}$, and $x, \xi$ distributed as above, define*

$$\mathbf{ENS}_\nu(\{f_w\}_{w \sim \mu}) := \mathbb{P}_{w \sim \mu, x, \xi}[f_w(x) \neq f_w(x \odot \xi)].$$

**Fourier analysis and spectral concentration.** Every $f : \{-1,1\}^n \to \mathbb{R}$ admits a unique Fourier–Walsh expansion (O'Donnell, 2014, Thm. 1.1):

$$f(x) = \sum_{S \subseteq [n]} \hat{f}(S)\, \chi_S(x), \quad \chi_S(x) = \prod_{i \in S} x_i.$$

Low-degree terms ($|S|$ small) correspond to *low frequencies*, and high-degree terms to *high frequencies*. A constant function, for instance, has spectrum supported on $\emptyset$. The notion of *spectrum concentration* formalizes when a function is "simple": most of its Fourier weight lies on low-degree terms, i.e., subsets of size $o(n)$. Formally (O'Donnell, 2014, Def. 3.1), $f$ is $\epsilon$-concentrated up to degree $k$ if

$$\sum_{S:|S|>k} \hat{f}(S)^2 \leq \epsilon.$$

**Definition 4** (Expected spectrum concentration). *Given a parametric family of functions $\{f_\theta : \{-1,1\}^n \to \mathbb{R}\}_{w \in \mathcal{W}}$, and a probability measure $\mu$ in $\mathcal{W}$, we say that $\{f_w\}_{w \in \mathcal{W}}$ has, in expectation under $\mu$, spectrum $\epsilon$-concentrated up-to degree $k$ if*

$$\mathbb{E}_{w \sim \mu} \left[ \sum_{\substack{S \subseteq [n] \\ |S| > k}} \hat{f}_w^2(S) \right] \leq \epsilon.$$

A key connection between noise stability and spectral concentration is given by (O'Donnell, 2014, Prop. 3.3), which states that for a Boolean function $f$, if we set $\epsilon = 3\,\mathbf{NS}_\nu(f)$, then the spectrum of $f$ is $\epsilon$-concentrated up to degree $1/\nu$. This result extends naturally to parametric families of functions by linearity, replacing $\mathbf{NS}_\nu(f)$ with $\mathbf{ENS}_\nu(f)$ and spectral concentration with its expected counterpart (see Lemma 6).

**Other notions of simplicity.** Alternative notions of simplicity have been proposed in the simplicity bias literature (Valle-Pérez et al., 2019; De Palma et al., 2019). Closest to our setting is the definition of De Palma et al. (2019), who study, for a classifier $f : \{-1,1\}^n \to \{-1,1\}$ and $x \in \{-1,1\}^n$, the quantity

$$N_h(x; f) := \big| \{ y : d_H(x,y) = h, \ f(x) \neq f(y) \} \big|, \tag{4}$$

which counts $h$-bit perturbations that flip the label. They show that its expected asymptotic behavior of depends on the activation function; for ReLU networks $\mathbb{E}_{x \sim \mathrm{Unif}(\{-1,1\}^n)}[N_h(x, f)] \to 0$, implying that the average Hamming distance to the nearest input with a different class is $\mathcal{O}(\frac{\sqrt{n}}{\log n})$. To connect this notion with noise sensitivity, note that for any parametric family of Boolean functions $\{f_w\}_{w \in \mathcal{W}}$,

$$\mathbf{ENS}_\nu\left(\{f_w\}_{w \sim \mu}\right) = \sum_{h=1}^n \mathbb{E}_{w \sim \mu, x \sim \mathrm{Unif}}[N_h(x; f_w)]\, \nu^h (1-\nu)^{n-h}.$$

In principle, one could recover $\mathbb{E}[N_h(x; f_w)]$ by inverting the previous relation, but this would require a precise characterization of $\mathbf{ENS}_\nu$. In practice, only bounds on this quantity are typically attainable, as we show next.

## 4 NOISE STABILITY OF SNN CLASSIFIERS

In this section, we apply the Boolean function analysis framework from Section 3 to quantify the stability of discrete-time LIF-SNN classifiers. Recalling the SNN classifier definition in equation 3 and our main assumptions from Section 2, we begin with the single neuron case.

**Single neuron stability.** Under Definition 1, each output $s_t$ in the sequence $(s_t)_{t \in [T]}$ defines a Boolean function $s_t : \{-1, 1\}^n \to \{-1, 1\}$. For any $t \in [T]$ and input sequences $(x_t)_{t \in [T]}$ and $(y_t)_{t \in [T]}$ with fixed Hamming distance, we bound the probability (over random initialization) that $s_t((x_k)_{k \in [t]}, w) \neq s_t((y_k)_{k \in [t]}, w)$.

**Theorem 1.** *Consider a sLIF neuron, with latency $T \in \mathbb{N}_+$, threshold $\theta \in (0, \infty)$, with random parameter vector $w \sim \mathcal{N}(0, I_n/n)$. We consider two input sequences $x_1, \ldots, x_T \in \{-1, 1\}^n$ and $y_1, \ldots, y_T \in \{-1, 1\}^n$. Denote $\nu_t = d_H(x_t, y_t)/n$ and $\overline{\nu}_t = \frac{1}{t} \sum_{k=1}^{t} \nu_t$. If $\max_{t \in [T]} \nu_t = \mathcal{O}(\frac{1}{\sqrt{n}})$, then for all $t \in [T]$, we have*

$$\mathbb{P}_w \left[ s_t(x_1, \ldots, x_t) \neq s_t(y_1, \ldots, y_t) \right] \leq C(1+\theta)t^2 \sqrt{\overline{\nu}_t} \log n,$$

*where $C > 0$ is an absolute constant independent of the $\theta, T, n, t, x_1, \ldots, x_T$ and $y_1, \ldots, y_T$. Moreover, for static inputs the same bound applies without the logarithmic factor.*

Below we present a proof sketch of Theorem 1 focusing on the primary technical challenges; the full proof is deferred to Appendix C.1. For the case with $\beta \neq 1$, and with Heaviside activation instead of sign, we refer the reader to Appendix D.

*Proof sketch of Theorem 1.* We argue by induction. Here, we illustrate the argument for $t = 1$. In this case, the problem reduces to bounding the probability that a random linear threshold function produces different outputs for two inputs $x_1, y_1$ at Hamming distance $\lfloor \nu_1 n \rfloor$:

$$\mathbb{P} \left[ \text{sign}(w^\top x_1 - \theta) \neq \text{sign}(w^\top y_1 - \theta) \right].$$

*Step 1: Decomposition.* Define $X = w^\top x_1$ and $Y = w^\top y_1$, which are standard Gaussians. A classic Gaussian decomposition property allow us to express $Y = \rho X + \sqrt{1 - \rho^2} Z$, where $\rho = 1 - \nu_1$.

*Step 2: Event characterization.* The disagreement event is equivalent to

$$\{X > \theta, Y \leq \theta\} \cup \{X \leq \theta, Y > \theta\}.$$

*Step 3: Probability estimate.* The first event has probability

$$\mathbb{P}[X > \theta, Y \leq \theta] = \Phi_2\left(-\theta, \theta; 2\nu_1 - 1\right),$$

where $\Phi_2$ denotes the bivariate Gaussian CDF with unit variances and correlation $2\nu_1 - 1$. The second event yields a symmetric expression $\Phi_2(\theta, -\theta; 2\nu_1 - 1)$. Combining both, and using tail bounds and Lemma 3, one obtains

$$\mathbb{P}[\text{sign}(w^\top x - \theta) \neq \text{sign}(w^\top y - \theta)] \leq C_\theta \sqrt{\nu_1},$$

for a constant $C_\theta$ depending only on $\theta$, see equation 10 for details.

For $t \geq 2$, the argument extends but with added complexity: temporal dependencies require union bounds, introducing the $T$ factor in the constant. In the dynamic case, the sLIF neuron processes input sums that lie outside the hypercube, creating technical challenges. In contrast, the static case avoids these issues and the proof is simpler, with better rates.

$\square$

**Remarks.** For $T = 1$, our result recovers (up to logarithmic factors) the known bounds on the noise sensitivity of fixed linear threshold functions, i.e., classifiers of the form $\text{sign}(w^\top x - \theta)$. For larger $T$, our bounds deteriorate. This may partly reflect proof artifacts—since handling dependencies introduced by shared weights across time can loosen bounds—but it is also consistent with the general behavior of Boolean function compositions, where sensitivity typically increases with depth.

A key obstacle to sharper time dependencies is the reset mechanism: thresholds adapt dynamically as the process evolves, complicating tighter analysis.

By inspecting the proof, we note that the assumption $\beta = 1$ can be relaxed without significant changes to the analysis, but we maintain it here for simplicity. The proof also applies, with minor modifications, to the classical LIF model with Heaviside activations, but we adopt the signed variant to enable a cleaner Fourier analysis. In contrast, the requirement of large network width is essential, as our concentration-based arguments rely on it. The influence of architectural parameters on stability is explored empirically in Section 5.

**Multiple neurons.** We now analyze the noise sensitivity of classifier outputs from $L$-layer sLIF neural networks. Specifically, we study classifiers defined as in equation 3 and built from $L$-layer sLIF networks (Definition 2).

**Theorem 2.** *Let $f^{L,T}(\cdot, W)$ be a $n_L$-classes classifier, defined by a $L$-layer sLIF, according to equation 3, with latency $T \in \mathbb{N}_+$, $\theta \in (0, \infty)$, widths $n_1 = n_2 = \ldots = n_{L-1} = n$ and weights $W \sim \mathcal{N}(0, I_d/n)$. Let $x_1, \ldots, x_T \in \{-1, 1\}^n$ and $y_1, \ldots, y_T \in \{-1, 1\}^n$ such that $d_H(x_t, y_t) = \lfloor \nu_t n \rfloor$ with $\nu_t \in [0, 1]$. Let us define $\nu := \max_{t \in [T]} \nu_t$ and assume $\nu = \mathcal{O}(\frac{1}{\sqrt{n}})$. Then, for $n$ large enough, it holds that*

$$\mathbb{P}_W \left( f^{L,T}((x_t)_{t \in [T]}, W) \neq f^{L,T}((y_t)_{t \in [T]}, W) \right)$$

$$\leq n_L T^4 C (1 + \theta) \nu^{\frac{1}{2^{2L+1}}} \log^{3/2} n + (L-1) e^{-c\nu^{\frac{1}{2^{2L-1}}} n},$$

*for some absolute constants $c, C > 0$ independent of $\theta$, $T$, $n$, $t$, $x_1, \ldots, x_T$ and $y_1, \ldots, y_T$.*

*Proof sketch of Theorem 2.* We proceed by induction on $l \in [L]$. To illustrate, consider $t = 1$. Following (Jonasson et al., 2023), we analyze the Markov chain

$$D_1^{(l)}(x_1, y_1) := \tfrac{1}{4} \| s_1^{(l)}(x_1) - s_1^{(l)}(y) \|^2, \qquad l \in [L],$$

which has $n + 1$ states and absorbing state 0. Conditioned on $D_1^{(l-1)} = \lfloor \nu_1 n \rfloor$, we have

$$D_1^{(l)}(x, y) \sim \text{Bin}(n, p_{\nu_1}), \quad p_{\nu_1} \leq C_\theta \sqrt{\nu_1},$$

where the bound on $p_{\nu_1}$ follows from Theorem 1. Hence $D_1^{(l)}(x, y)$ is stochastically dominated by $\text{Bin}(n, C_\theta \sqrt{\nu_1})$, which leads to the desired bound via Chernoff bounds (see Theorem 3) and standard manipulations. For $t \geq 2$, we repeat the previous argument which involves repeatedly applying Chernoff bounds ; see Appendix D.3. $\square$

The following corollary links the probability that input perturbations change the output to a bound on the expected noise sensitivity of binary sLIF-SNN classifiers. This, in turn, characterizes their spectrum and quantifies the degree of spectral simplicity introduced in Section 3. The proof is deferred to Appendix D.4.

**Corollary 1.** *Let $f^{L,T}(\cdot, W)$ be a binary classifier ($n_L = 2$) as in Theorem 2, and assume static inputs. Then, for any $\nu' \leq \frac{1}{\sqrt{n} \log n}$, we have, for $n$ large enough,*

$$\mathbf{ENS}_{\nu'} \left( \{f^{L,T}(\cdot, W)\}_{W \sim \mathcal{N}(0, I_d)} \right) \leq C_{T,\theta} \nu'^{\frac{1}{2^{2L+1}}} \log^{3/2} n + (L-1) e^{-c\nu'^{\frac{1}{2^{2L-1}}} n} + e^{-\frac{1}{4}\sqrt{n}},$$

*where $C_{T,\theta} > 0$ is a constant, as in Theorem 2. Moreover, for sufficiently large $n$, the family of $L$-layer sLIF SNN binary classifiers has, in expectation under $\mathcal{N}(0, I_d/n)$, spectrum $\epsilon$-concentrated (Definition 4) up to degree $1/\nu'$, with*

$$\epsilon = C_{T,\theta} \nu'^{\frac{1}{2^{2L+1}}} \log^{3/2} n.$$

**Remarks.** To illustrate, take $\nu' = \frac{1}{\sqrt{n} \log n}$. In this case, an $L$-layer binary SNN classifier is $\mathcal{O}(n^{1/2^{2(L+1)}})$-concentrated up to degree $\mathcal{O}(\sqrt{n} \log n)$. Thus, only a vanishing fraction of degrees contribute meaningfully to the spectrum, making these classifiers spectrally simple. Interestingly, the bound on the maximal degree of concentration is independent of architectural parameters, while

the concentration level deteriorates with larger $L$, $T$, and $\theta$ (the $\log^{3/2} n$ seems an artifact of the proof). The increase with $L$ is not surprising, since compositions of Boolean networks typically behave similarly, and analogous results are known for threshold ANNs (Jonasson et al., 2023). The $\theta$-dependence appears to be a proof artifact. Whether the $T$- and $L$-dependencies are intrinsic remains an open question, which we investigate experimentally in the next section.

## 5 NUMERICAL EXPERIMENTS

We empirically evaluate the noise sensitivity, $\mathbf{ENS}_\nu$, of various spiking neural networks to investigate how our proposed simplicity measure relates to the model dimension $n$ and the tightness of the bounds established in Theorems 1 and 2 in the case of static inputs. In addition, we study how training (signed) SNNs influences their sensitivity to random perturbations of the input.

**Single LIF experiments.** We approximate the noise sensitivity $\mathbf{ENS}_{1/\sqrt{n}}$ for different IF and sIF spiking neurons with input dimensions $n = 100, 1000, 10000$, threshold $\theta = 0.5$ and $T = 10$. Specifically, we compute a Monte Carlo approximation by uniformly sampling 10 model's weights, 100 data points, and 100 random perturbations. The results are shown in Figure 1a and 1b. We observe that, for all considered $t$, the neurons exhibit low sensitivity for both sIF and IF models. In conclusion, we stress that even though for our result we need the signed neuron versions, the sensitivity seems to be similar for the IF neuron. Moreover, the theoretical bound proved in Theorem 1 is satisfied both for the sIF and the IF neuron.

Additional experiments for different values of $\nu$, $\beta = 0.5$, and $\theta = 0$ can be found in Appendix E, see Figure 5, Figure 6, and Figure 7 respectively.

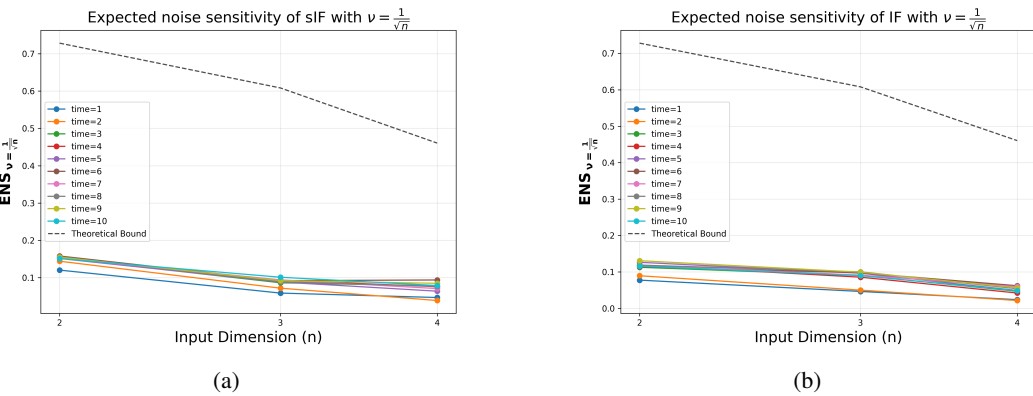

Figure 1: Noise sensitivity $\mathbf{ENS}_{1/\sqrt{n}}$ for different input dimensions $n$ for sIF and IF neurons with $\theta = 0.5$ and $T = 10$. **(a)** sIF neuron (log-scale x-axis); dashed line: scaled bound from Theorem 1. **(b)** IF neuron (log-scale x-axis); dashed line: scaled bound from Theorem 1.

**IF SNN with 5 layers.** We extend the experiments to the deep setting by considering IF and sIF spiking neural networks with five layers, using the same Monte Carlo approximation procedure as in the shallow case. We evaluate $\mathbf{ENS}_{1/\sqrt{n}}$ for input dimensions $n = 100, 1000, 10000$, with each layer having width equal to the input dimension, $\theta = 0.5$ and $T = 10$. The results are shown in Figure 2a and 2b. Although depth appears to have a stronger impact on sensitivity than latency, the bound presented in Theorem 2 tends to overestimate the effect. Additional experiments for different values of $\nu$ and $\beta = 0.5$ can be found in Appendix E.

**Noise sensitivity after training.** We evaluate the noise sensitivity of trained sIF and IF SNNs in static data. In particular, we train three-layer sLIF and IF SNN on MNIST (i.e., $n = 784$) using the ADAM optimizer with surrogate gradients (Eshraghian et al., 2023) until reaching $98\%$ training accuracy. After training, we estimate the output sensitivity by perturbing each test sample 100 times, flipping each component with probability $\nu \in \left[\frac{1}{n}, \frac{2}{n} \ldots, \frac{1}{\sqrt{n}}\right]$ and measuring the fraction of perturbations that change the network's output. For comparison with a random network, the noise

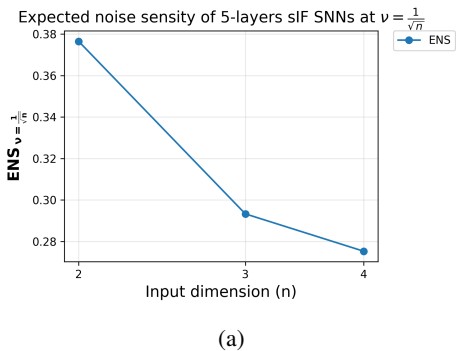 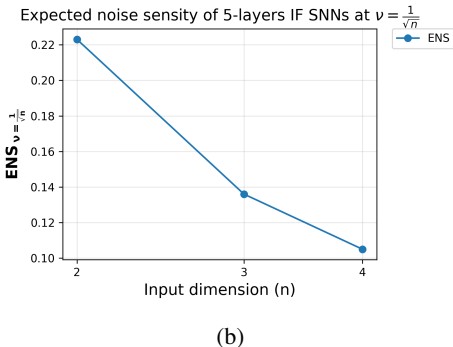

(a)  (b)

Figure 2: Noise sensitivity $\mathbf{ENS}_{1/\sqrt{n}}$ for different input dimensions $n$ for 5-layers sIF and IF neural networks with $\theta = 0.5$ and $T = 10$. **(a)** sIF neuron (log-scale x-axis); **(b)** IF neuron (log-scale x-axis).

sensitivity is estimated using the same procedure, averaged over 10 independent random weight initializations. As expected, training significantly reduces the sensitivity of the model whenever the final test accuracy is sufficiently high. Figure 3a shows show the results for MNIST. Figure 3 shows the same experiment on CIFAR-10 ($n = 3072$). Notice that training reduces the model's sensitivity, but less strongly compared to MNIST. This aligns with the fact that both the training and test accuracies are larger for CIFAR-10 (which achieves $84.38\%$ training accuracy).

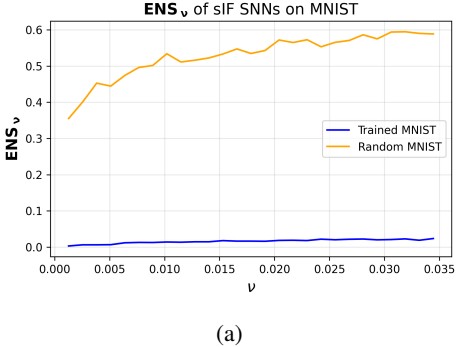 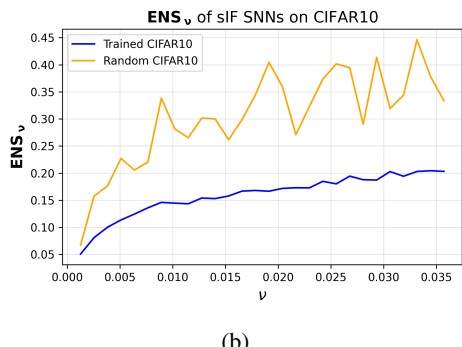

(a)  (b)

Figure 3: Sensitivity to input perturbations in sLIF-SNNs ($T = 100$, $\theta = 0.5$, $\beta = 1$, $L = 3$), shown at initialization and after training on **(a)** MNIST and **(b)** CIFAR-10.

**Noise sensitivity on neuromorphic dataset.** We evaluate the noise sensitivity of a spiking convolutional network on NMNIST Orchard et al. (2015). NMNIST is an event-based version of MNIST recorded with a dynamic vision sensor. Each data point is a tensor in $\{0, 1\}^{T \times 2 \times 34 \times 34}$ where $T \approx 30$ is the time, $2$ is the polarity and $34$ are the width and the height. We perturb the input sequence by replacing each time frame by flipping each component with probability $\nu \in \left[\frac{1}{n}, \frac{2}{n} \dots, \frac{1}{\sqrt{n}}\right]$ where $n = 2312 (= 34 \times 34 \times 2)$. The results are displayed in Figure 4. Notice that the model has small ENS both before and after training. Moreover, the training does not affect the ENS as much as for static data.

**Dropping 5% of the input.** We evaluate the noise sensitivity when the 5% of the input component are randomly dropped. In detail, we evaluate the noise sensitivity for a 5-layers (s)SNN with input dimension $n = 1000$, and we compare the probability of observing different output with respect to this input perturbation. The results are reported in Table 1. We notice that, both for the LIF and the sLIF model, dropout leads to smaller output sensitivity than random flipping. This is indeed expected since dropout does not affect zero components in the case of inputs in the $\{0, 1\}^n$ hypercube.

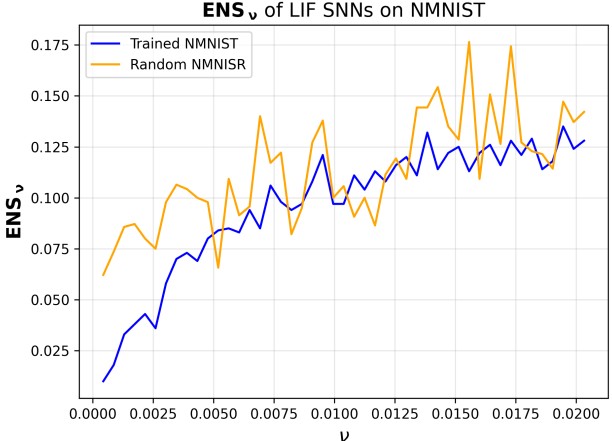

Figure 4: Sensitivity to input perturbations of a convolutional SNN with $T = 100$, $\theta = 0.5$ and $\beta = 0.5$, shown at initialization and after training on NMNIST

Table 1: Sensitivity of 5-layer (s)LIF neural networks with $\beta = 0.5$ and $\theta = 1$ under two types of input perturbations: *random flipping* and *dropout*. For each model, the lowest (i.e., most robust) sensitivity value is highlighted in bold.

| Model / Perturbation | Random Flipping | Dropout |
|---|---|---|
| sLIF | 0.19 | **0.16** |
| LIF | 0.28 | **0.16** |

## 6 CONCLUSION

In this paper, we study the stability of wide SNN classifiers through the lens of Boolean function analysis. We provide quantitative bounds on their expected noise sensitivity and show how these stability guarantees connect to simplicity bias, motivating a new notion of simplicity. We empirically validate Theorem 1 in the case of single sIF and IF neuron. We show that both shallow and deep (s)IF neural networks exhibit a small noise sensitivity in practice, and this property extends also to random dropout perturbation of the input signal. Furthermore, we empirically suggest that training tends to preserve or even improve the ESN of SNNs.

The classifiers we analyze are widely used in practice, and most of our assumptions can be relaxed. The main restriction is the requirement of sufficiently large widths, needed to apply concentration of measure; whether this condition can be weakened remains an open question. On the other hand, the uniform input distribution assumed in Definition 3 does not affect Corollary 1, making extensions to other input distributions straightforward.

**Future directions.** Several avenues merit further investigation:

- Extending our results to feedforward RNNs, where the dynamics are simpler, as well as to more general non-feedforward SNNs.
- Studying stability under alternative perturbation distributions, a step toward understanding adversarial robustness.
- Investigating the average distance to the nearest input with a different label (see equation 4), as in (De Palma et al., 2019). Unlike their setting, a simple union bound fails here, since our $\mathcal{O}(1/\sqrt{n})$ bound is insufficient given the exponentially many inputs at distance $\mathcal{O}(\sqrt{n})$.
- Understanding how initialization impacts stability in SNNs, and whether classical ANN initialization schemes are optimal in this context.

## ACKNOWLEDGEMENTS

Ernesto Araya, Massimiliano Datres and Gitta Kutyniok acknowledge support by the Munich Center for Machine Learning (MCML).

Ernesto Araya and Gitta Kutyniok acknowledge support by the German Research Foundation under Grant DFG-SPP-2298.

Massimiliano Datres and Gitta Kutyniok acknowledge support by the project "Next Generation AI Computing (gAIn)", funded by the Bavarian Ministry of Science and the Arts and the Saxon Ministry for Science, Culture, and Tourism as well as by the Hightech Agenda Bavaria.

Gitta Kutyniok is also grateful for partial support from the Konrad Zuse School of Excellence in Reliable AI (DAAD). Furthermore, she acknowledges support by the German Research Foundation under Grants KU 1446/31-1 and KU 1446/32-1.

## ETHICS STATEMENT

This work focuses on the theoretical analysis of robustness in machine learning and does not involve experiments on human subjects, sensitive personal data, or applications with direct societal risks. The datasets referenced are publicly available, and no private or restricted data was used. Potential ethical concerns related to misuse are minimal, as the contributions are mainly theoretical.

**Acknowledgment of LLM Use.** We explicitly acknowledge that large language models (LLMs) were used solely for polishing code, improving sentence clarity, and refining grammar. They were not used for generating research ideas, proofs, or results.

## REPRODUCIBILITY STATEMENT

We have taken multiple steps to ensure reproducibility of our results. All theoretical claims are accompanied by rigorous proofs, presented in detail in the appendix. Assumptions underlying the theorems are explicitly stated, and definitions are given in full to allow independent verification.

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

## A   KNOWN RESULTS IN STATISTICS

In this appendix, we recall some standard results from probability and statistics that are used throughout our proofs

**Lemma 1** (Linear Combination of Gaussians). *Let $X_1, \ldots, X_n$ be independent Gaussian random variables with $X_i \sim \mathcal{N}(\mu_i, \sigma_i^2)$, and let $a_1, \ldots, a_n \in \mathbb{R}$. Then the linear combination*

$$Y = \sum_{i=1}^{n} a_i X_i$$

*is also a Gaussian random variable with mean and variance given by:*

$$\mathbb{E}[Y] = \sum_{i=1}^{n} a_i \mu_i, \quad \mathrm{Var}(Y) = \sum_{i=1}^{n} a_i^2 \sigma_i^2.$$

**Theorem 3** (Chernoof Bound). *Let $\{X_i\}_{i=1}^{n}$ a sequence of independent random variables such that $X_i = 1$ with probability $p_i$ and $X_i = 0$ with probability $1 - p_i$. Let us consider $X = \sum_{i=1}^{n} X_i$. Then $\mu := \mathbb{E}[X] = \sum_{i=1}^{n} p_i$ and for all $\varepsilon > 0$:*

$$\mathbb{P}(X \geq (1 + \varepsilon)\mu) \leq e^{-\frac{\varepsilon^2}{2+\varepsilon}\mu}$$

**Definition 5.** *Let $X$ and $Y$ be two random variables such that*

$$P\{X > x\} \leq P\{Y > x\} \quad \text{for all } x \in (-\infty, \infty)$$

*then $X$ is said to be smaller than $Y$ in the usual stochastic order (denoted by $X <_{\mathrm{st}} Y$ ).*

We have the following result for the stochastic domination in Binomial variables. We include its proof for completeness.

**Lemma 2.** *If $X \sim Bin(n, p)$ and $Y \sim Bin(n, q)$ for some $0 < p < q < 1$. Then, for all $k \in [0, n]$, we have that $X <_{\mathrm{st}} Y$.*

*Proof.* Let us recall that the probability mass functions:

$$f_X(k) = \binom{n}{k} p^k (1-p)^{n-k}, \quad f_Y(k) = \binom{n}{k} q^k (1-q)^{n-k},$$

and, hence, we have

$$\frac{f_Y(k)}{f_X(k)} = \left(\frac{q}{p}\right)^k \left(\frac{1-q}{1-p}\right)^{n-k} =: \Phi(k).$$

Let $h(k) := f_X(k) - f_Y(k)$. Since $f_X$ and $f_Y$ are both probability mass functions, it holds $\sum_k h(k) = \sum_k (f_X(k) - f_Y(k)) = 0$ and, hence, $h$ is a signed measure with total mass zero. Since the $\Phi$ is increasing in $k$, $\Phi(0) < 1$ and $\Phi(n) > 1$, by continuity of $\Phi$, there exists a (unique) $k^* \in (0, n)$ such that:

1. $0 < \Phi(k) < 1$ for all $k \in (0, k^*)$;

2. $\Phi(k) > 1$ for all $k \in (k^*, n]$.

Therefore, we have that:

$$h(k) > 0 \text{ for } k \in (0, k^*) \text{ and } h(k) < 0 \text{ for } k \in (k^*, n] \tag{5}$$

Combining equation 5 and $\sum_{k=0}^{n} h(k) = 0$, we notice that $\sum_{j=0}^{k} h(j) \geq 0$ for all $k \in [n]$. Therefore, for all $k \in [0, n]$, we obtain that

$$\mathbb{P}(X > k) = 1 - \sum_{j=0}^{k} f_X(j) = 1 - \sum_{j=0}^{k} f_Y(j) - \sum_{j=0}^{k} (f_X(j) - f_Y(j)) = 1 - \sum_{j=0}^{k} f_Y(j) - \sum_{j=0}^{k} h(j)$$

$$\leq 1 - \sum_{j=0}^{k} f_Y(j) = \mathbb{P}(Y > k),$$

which concludes the proof. □

The following result may already be established; however, since we were unable to find a reference, we provide a proof here.

**Lemma 3.** *Let $\rho \in (0, 1], a, b > \mathbb{R}$ and $\overline{X}, \overline{Z} \overset{i.i.d.}{\sim} \mathcal{N}(0,1)$, it holds*

$$\mathbb{P}[\overline{X} \le a,\ \rho\overline{X} + \sqrt{1-\rho^2}Z > b] \le \sqrt{1-\rho^2} + |a|\frac{1-\rho}{\rho} + \left|\frac{(b-a)}{\rho}\right|$$

$$\mathbb{P}[\overline{X} > a,\ \rho\overline{X} + \sqrt{1-\rho^2}Z \le b] \le \sqrt{1-\rho^2} + |a|\frac{1-\rho}{\rho} + \left|\frac{(b-a)}{\rho}\right|.$$

*Proof.* Let $\varphi(x) := \frac{1}{\sqrt{2\pi}}e^{-x^2/2}$ be the density of a normal random variable and $\Phi(x)$ the corresponding CDF, then

$$\mathbb{P}[\overline{X} > a,\ \rho\overline{X} + \sqrt{1-\rho^2}Z \le b] = \int_a^\infty \Phi\left(\frac{b - \rho x}{\sqrt{1-\rho^2}}\right)\varphi(x)dx$$

$$= \int_{-\infty}^{\frac{b-\rho a}{\sqrt{1-\rho^2}}} \Phi(\eta)\varphi\left(\frac{\sqrt{1-\rho^2}\eta - b}{\rho}\right)\frac{\sqrt{1-\rho^2}}{\rho}d\eta$$

$$\le \int_{-\infty}^0 \Phi(\eta)\varphi\left(\frac{\sqrt{1-\rho^2}\eta - b}{\rho}\right)\frac{\sqrt{1-\rho^2}}{\rho}d\eta + \left|\int_0^{\frac{b-\rho a}{\sqrt{1-\rho^2}}} \Phi(\eta)\varphi\left(\frac{\sqrt{1-\rho^2}\eta - b}{\rho}\right)\frac{\sqrt{1-\rho^2}}{\rho}d\eta\right|$$

The first term can be bounded by

$$\int_{-\infty}^0 \Phi(\eta)\varphi\left(\frac{\sqrt{1-\rho^2}\eta - b}{\rho}\right)\frac{\sqrt{1-\rho^2}}{\rho}d\eta \le \int_{-\infty}^0 \frac{1}{\sqrt{2\pi}}e^{-\frac{\eta^2}{2} - \frac{(\sqrt{1-\rho^2}\eta - b)^2}{2\rho^2}}\frac{\sqrt{1-\rho^2}}{\rho}d\eta$$

$$= \frac{\sqrt{1-\rho^2}}{\sqrt{2\pi}\rho}\int_{-\infty}^0 e^{-\frac{\eta^2\rho^2 + (1-\rho^2)\eta^2 - 2\sqrt{1-\rho^2}\eta b + b^2}{2\rho^2}}d\eta$$

$$= \frac{\sqrt{1-\rho^2}}{\sqrt{2\pi}\rho}\int_{-\infty}^0 e^{-\frac{\eta^2 - 2\sqrt{1-\rho^2}\eta b + b^2}{2\rho^2}}d\eta$$

$$= \frac{\sqrt{1-\rho^2}}{\sqrt{2\pi}\rho}e^{-\rho^2 b^2}\int_{-\infty}^0 e^{-\frac{(\eta - \sqrt{1-\rho^2}b)^2}{2\rho^2}}d\eta$$

$$\le \sqrt{1-\rho^2}$$

We control now the second term, obtaining that

$$\left|\int_0^{\frac{b-\rho a}{\sqrt{1-\rho^2}}} \Phi(\eta)\varphi\left(\frac{\sqrt{1-\rho^2}\eta - b}{\rho}\right)\frac{\sqrt{1-\rho^2}}{\rho}d\eta\right|$$

$$\le \left|\frac{b - \rho a}{\sqrt{1-\rho^2}}\right|\frac{\sqrt{1-\rho^2}}{\rho}$$

$$= \left|\sqrt{\frac{1-\rho}{1+\rho}}a + \frac{(b-a)}{\sqrt{1-\rho^2}}\right|\frac{\sqrt{1-\rho^2}}{\rho}$$

$$\le |a|\left|\frac{\sqrt{(1-\rho)(1-\rho^2)}}{\sqrt{(1+\rho)}\rho}\right| + \left|\frac{(b-a)}{\sqrt{1-\rho^2}}\right|\frac{\sqrt{1-\rho^2}}{\rho}$$

$$\le |a|\frac{1-\rho}{\rho} + \left|\frac{(b-a)}{\rho}\right|$$

Combining the previous equations, we conclude the proof by noticing that

$$\mathbb{P}[\overline{X} > a,\ \rho\overline{X} + \sqrt{1-\rho^2}Z \le b] = \mathbb{P}[-\overline{X} \le -a,\ -\rho\overline{X} - \sqrt{1-\rho^2}Z > -b]$$

and $-\overline{X}$ and $-\rho\overline{X} - \sqrt{1-\rho^2}Z$ are $\rho$-correlated. $\qquad\square$

## B TECHNICAL LEMMAS

We now provide a more explicit expression for the output of a sLIF neuron with threshold $\theta$ at time $t$. We give the result with $\beta = 1$, but it clearly generalizes to $\beta \in [0, 1]$ by considering a weighted sum of the input sequence, instead of the sum itself.

**Lemma 4.** *Let $\theta > 0, T \in \mathbb{N}_+$, and $w \in \mathbb{R}^n$. Let us consider a (discrete) sLIF neuron, according to equation 1, with parameters $w, \theta, T$ and $\beta = 1$, and input signal $(x_t)_{t \in [T]}$. Then, for all $t \in [T]$, the output of the sLIF neuron at time $t$ can be computed recursively as described below*

$$\begin{cases} s_t = \text{sign}\left(w^\top \left(\sum_{k=1}^t x_k\right) - \theta\left(1 + \frac{1}{2}\sum_{k=0}^{t-1}(s_k + 1)\right)\right) \\ s_0 = -1 \end{cases}.$$

*Proof.* For $k \in [T]$, we notice that

$$u_k - u_{k-1} = w^\top x_k - \frac{\theta}{2}(s_{k-1} + 1),$$

then, summing over $k \in [t]$, we get

$$u_t = w^\top\left(\sum_{k=1}^t x_k\right) - \frac{\theta}{2}\sum_{i=1}^t (s_{i-1} + 1),$$

from which, together with equation 1, the result follows. $\square$

In the next lemma, we address a key challenge for dynamic inputs: although each element of an input sequence $x_1, \ldots, x_T$ lies in the hypercube $\{-1, 1\}^n$, their partial sums—processed by the neuron at each time step (cf. Lemma 4)—do not. This technical issue is absent in the static case, where the lemma is unnecessary.

**Lemma 5.** *Let $T \geq 1$, $x_1, \ldots, x_T \in \{-1, 1\}^n$, $y_1, \ldots, y_T \in \{-1, 1\}^n$ and define $h_t = d_H(x_t, y_t)$, $h = \frac{1}{T}\sum_{t=1}^T h_t$, $\overline{x}_T := \frac{1}{T}\sum_{t=1}^T x_t$ and $\overline{y}_T := \frac{1}{T}\sum_{t=1}^T y_t$. Let us assume that $h = \mathcal{O}(\sqrt{n})$, then, it holds that either $\|\overline{y}_T\| = \|\overline{x}_T\| = \omega\left(\frac{\sqrt{n}}{\log n}\right)$ or $\|\overline{y}_T\| = \|\overline{x}_T\| = \mathcal{O}(\frac{\sqrt{n}}{\log n})$.*

*Proof.* Let $I_t := \{i \in [n] : x_{t,i} \neq y_{t,i}\}$ and $I = \bigcup_{t=1}^T I_t$. We notice that $h_t = |I_t|$ since $x_t, y_t \in \{-1, 1\}^n$, and therefore

$$|I| \leq \sum_{t=1}^T |I_t| = \sum_{t=1}^T h_t = hT. \tag{6}$$

We observe that

(i) $\overline{x}_T^{I^c} = \overline{y}_T^{I^c}$ by definition of $I$;

(ii) $\|\overline{x}_T\| = \|\overline{x}_T^I\| + \|\overline{x}_T^{I^c}\|$ and $\|\overline{y}_T\| = \|\overline{y}_T^I\| + \|\overline{y}_T^{I^c}\|$ since $I \cap I^c = \emptyset$.

Let us assume that $\|\overline{y}_T\| = \omega\left(\frac{\sqrt{n}}{\log n}\right)$. Then, we have two possibilities:

1. If $\|\overline{y}_T^{I^c}\| = \omega\left(\frac{\sqrt{n}}{\log n}\right)$, then $\|\overline{x}_T^{I^c}\| \overset{(i)}{=} \|\overline{y}_T^{I^c}\| = \omega\left(\frac{\sqrt{n}}{\log n}\right)$ and, using (ii), we conclude that $\|\overline{x}_T\| = \omega\left(\frac{\sqrt{n}}{\log n}\right)$;

2. If $\|\overline{y}_T^I\| = \omega\left(\frac{\sqrt{n}}{\log n}\right)$, combining equation 6 and the fact $|\overline{y}_{T,i}^I| \leq 1$ for $i \in I$ and $\overline{y}_{T,i}^I = 0$ for $i \in I^C$, we obtain that

$$hT > |I| \geq \|\overline{y}_T^I\|^2 = \omega\left(\frac{n}{\log^2 n}\right).$$

We note that, since $h = \mathcal{O}(\sqrt{n})$, this scenario cannot occur for $n$ large enough.

Finally, we conclude that either $\|\overline{y}_T\| = \|\overline{x}_T\| = \omega\left(\frac{\sqrt{n}}{\log n}\right)$ or $\|\overline{y}_T\| = \|\overline{x}_T\| = \mathcal{O}(\frac{\sqrt{n}}{\log n})$ $\qquad\square$

We now give a straightforward generalization of (O'Donnell, 2014, Prop.3.3.) to the case of expected spectrum concentration (Definition 4).

**Lemma 6.** *For a parametric family* $\{f_w\}_{w \in \mathcal{W}}$, *a probability measure* $\mu$ *on* $\mathcal{W}$, *and* $x \sim$ Unif$(\{-1,1\}^n)$, $\xi \sim \mathrm{Rad}(1-\nu)$. *Then, for* $\nu \in (0, 1/2]$,

$$\mathbb{E}_{w \sim \mu}\left[\sum_{\substack{S \subseteq [n] \\ |S| > 1/\nu}} \hat{f}_w^2(S)\right] \leq 4\,\mathbf{ENS}_\nu\left(\{f_w\}_{w \sim \mu}\right).$$

*Proof.* Following (O'Donnell, 2014), for a fixed $f_w$,

$$\sum_{\substack{S \subseteq [n] \\ |S| > 1/\nu}} \hat{f}_w^2(S) = \mathbb{P}_{S \sim \mathbf{S}_{f_w}}\left[|S| > 1/\nu\right],$$

where the $\mathbf{S}_{f_w}$ denotes the probability distribution over the subsets of $[n]$, which assigns probability $\hat{f}_w^2(S)$ to the subset $S$ (recall that the Fourier coefficients sum to 1). Then, using (O'Donnell, 2014, Thm.2.49)

$$2\,\mathbf{NS}_\nu(f_w) = \mathbb{E}_{S \sim \mathbf{S}_{f_w}}\left[1 - (1 - 2\nu)^{|S|}\right]$$
$$\geq \left(1 - (1 - 2\nu)^{1/\nu}\right)\mathbb{P}_{S \sim \mathbf{S}_{f_w}}\left[|S| > 1/\nu\right]$$
$$\geq \frac{1}{2}\sum_{\substack{S \subseteq [n] \\ |S| > 1/\nu}} \hat{f}_w^2(S).$$

In the first inequality, the fact that $1 - (1 - 2\nu)^k$ is non-decreasing in $k$ is used. Then the result follows by taking expectation with respect to $w \sim \mu$. $\qquad\square$

## C   PROOFS OF THE MAIN RESULTS

### C.1   PROOF OF THEOREM 1

Consider $x_1, \ldots, x_T \in \{-1, 1\}^n$ and $y_1, \ldots, y_T \in \{-1, 1\}^n$ with $d_H(x_t, y_t) = h_t$ for $t \in [T]$ and define $\overline{h}_t = \frac{1}{t}\sum_{k=1}^t h_k$. Notice that $h_t = \lfloor \nu_t n \rfloor$ and $\overline{h}_t = \lfloor \overline{\nu}_t n \rfloor$. We proceed by induction over $t \in [T]$.

The base case $t = 1$ requires to control $\mathbb{P}\left[s_1(x_1, w) \neq s_1(y_1, w)\right]$. We note that $\overline{h}_1 = h_1$. Let us define $X = \sqrt{n}w^\top \frac{x_1}{\|x_1\|}$ and $Y = \sqrt{n}w^\top \frac{y_1}{\|y_1\|}$, then $X$ and $Y$ are $\rho$-correlated with

$$\rho = \left\langle \sqrt{n}w^\top \frac{x_1}{\|x_1\|}, \sqrt{n}w^\top \frac{y_1}{\|y_1\|}\right\rangle = \frac{n}{\|x_1\|\|y_1\|}\sum_{j=1}^n w_j^2 x_i y_i$$
$$\geq \frac{1}{2}\left(\frac{\|x_1\|}{\|y_1\|} + \frac{\|y_1\|}{\|x_1\|} - \frac{h_1}{\|x_1\|\|y_1\|}\right) \qquad (7)$$
$$= \frac{1}{2}\left(\frac{\|x_1\|^2 + \|y_1\|^2 - h_1}{\|x_1\|\|y_1\|}\right)$$
$$\geq \left(1 - \frac{h_1}{2\|x_1\|\|y_1\|}\right).$$

Hence, it holds that

$$
\begin{aligned}
&\mathbb{P}\big[s_1(x_1, w) \neq s_1(y_1, w)\big]\\
&\quad = \mathbb{P}\big[\operatorname{sign}(w^\top x_1 - \theta) \neq \operatorname{sign}(w^\top y_1 - \theta)\big]\\
&\quad = \mathbb{P}\left[X > \theta\frac{\sqrt{n}}{\|x_1\|},\ Y \leq \theta\frac{\sqrt{n}}{\|y_1\|}\right] + \mathbb{P}\left[X \leq \theta\frac{\sqrt{n}}{\|x_1\|},\ Y > \theta\frac{\sqrt{n}}{\|y_1\|}\right].
\end{aligned}
\tag{8}
$$

Using Lemma 5, we can distinguish two cases

1. In the case $\|x_1\| = \|y_1\| = \mathcal{O}(\frac{\sqrt{n}}{\log n})$, we get

$$
\mathbb{P}\left[X > \theta\frac{\sqrt{n}}{\|x_1\|},\ Y \leq \theta\frac{\sqrt{n}}{\|y_1\|}\right] \leq \mathbb{P}\left[X > \theta\frac{\sqrt{n}}{\|x_1\|}\right] \overset{(i)}{\leq} e^{-\theta^2 \frac{n}{\|x_1\|^2}} = \mathcal{O}(e^{-\theta^2 \log^2 n}) \tag{9}
$$

$$
\mathbb{P}\left[X \leq \theta\frac{\sqrt{n}}{\|x_1\|},\ Y > \theta\frac{\sqrt{n}}{\|y_1\|}\right] \leq \mathbb{P}\left[Y > \theta\frac{\sqrt{n}}{\|y_1\|}\right] \overset{(ii)}{\leq} e^{-\theta^2 \frac{n}{\|y_1\|^2}} = \mathcal{O}(e^{-\theta^2 \log^2 n}),
$$

where in (i) and (ii) we used Gaussian tail bounds. Combining equation 9 and equation 8, it holds

$$
\mathbb{P}\big[s_1(x_1, w) \neq s_1(y_1, w)\big] = \mathcal{O}(e^{-\theta^2 \log^2 n}).
$$

2. In the case $\|x_1\| = \|y_1\| = \omega(\frac{\sqrt{n}}{\log n})$, combining equation 7 and equation 8, it holds

$$
\begin{aligned}
&\mathbb{P}\big[s_1(x_1, w) \neq s_1(y_1, w)\big]\\
&\quad = \mathbb{P}\left[X > \theta\frac{\sqrt{n}}{\|x_1\|},\ Y \leq \theta\frac{\sqrt{n}}{\|y_1\|}\right] + \mathbb{P}\left[X \leq \theta\frac{\sqrt{n}}{\|x_1\|},\ Y > \theta\frac{\sqrt{n}}{\|y_1\|}\right]\\
&\quad \overset{(iii)}{\leq} 2\sqrt{1 - \rho^2} + 2\theta\frac{\sqrt{n}}{\|x_1\|}\frac{1 - \rho}{\rho} + 2\theta\sqrt{n}\frac{|\|y_1\| - \|x_1\||}{\|x_1\|\|y_1\|\rho}\\
&\quad = 2\sqrt{1 + \rho}\sqrt{1 - \rho} + 2\theta\frac{\sqrt{n}}{\|x_1\|}\frac{1 - \rho}{\rho} + 2\theta\sqrt{n}\frac{|\|y_1\| - \|x_1\||}{\|x_1\|\|y_1\|\rho}\\
&\quad \overset{(iv)}{\leq} 2\sqrt{\frac{h_1}{\|x_1\|\|y_1\|}} + 2\theta\frac{\sqrt{n}}{\|x_1\|}\frac{h_1}{2\|x_1\|\|y_1\| - h_1} + 4\theta\frac{\sqrt{n}\sqrt{h_1}}{2\|x_1\|\|y_1\| - h_1}\\
&\quad \leq C\sqrt{\frac{h_1 \log^2 n}{n}} + C\theta\frac{h_1 \log^2 n}{n} + 4\theta C\sqrt{\frac{h_1 \log^2 n}{n}}\\
&\quad \leq C(1 + 5\theta)\sqrt{\frac{h_1 \log^2 n}{n}},
\end{aligned}
\tag{10}
$$

where $C > 0$ denotes a positive absolute constant independent by the model but dependent on the data, in (iii) we have used Lemma 3 and in (iv) we applied the inverse triangular inequality and the fact that $0 \leq \rho \leq 1$ since $h_1 = \mathcal{O}(\frac{n}{\log^2 n})$.

Combining equation 9 and equation 10, we conclude the proof of the base case, that is

$$
\mathbb{P}\big[s_1(x_1, w) \neq s_1(y_1, w)\big] \leq C(1 + 5\theta)\sqrt{\frac{h_1 \log^2 n}{n}},
$$

for $n$ large enough such that

$$
e^{-\theta^2 \log^2 n} < \sqrt{\frac{h_1 \log^2 n}{n}}.
$$

Let us now inductively assume that, for all $t \leq T - 1$, the following probability estimate holds

$$
\mathbb{P}[s_t(x) \neq s_t(y)] \leq C(1 + 5\theta)\, t\sqrt{\frac{\overline{h}_t \log^2 n}{n}}. \tag{11}
$$

Then, we have

$$\mathbb{P}[s_T(x) \neq s_T(y)]$$

$$= \mathbb{P}\left[s_T(x) \neq s_T(y), \sum_{t=1}^{T-1} s_t(x) \neq \sum_{t=1}^{T-1} s_t(y)\right] + \mathbb{P}\left[s_T(x) \neq s_T(y), \sum_{t=1}^{T-1} s_t(x) = \sum_{t=1}^{T-1} s_t(y)\right]$$

$$\leq \mathbb{P}\left[\sum_{t=1}^{T-1} s_t(x) \neq \sum_{t=1}^{T-1} s_t(y)\right] + \mathbb{P}\left[s_T(x) \neq s_T(y), \sum_{t=1}^{T-1} s_t(x) = \sum_{t=1}^{T-1} s_t(y)\right]$$

$$\leq \sum_{t=1}^{T-1} \mathbb{P}\left[s_t(x) \neq s_t(y)\right] + \mathbb{P}\left[s_T(x) \neq s_T(y), \sum_{t=1}^{T-1} s_t(x) = \sum_{t=1}^{T-1} s_t(y)\right] \quad (12)$$

$$\overset{(v)}{\leq} \sum_{t=1}^{T-1} C(1 + 5\theta)\, t \sqrt{\frac{\overline{h}_t}{n}} + \mathbb{P}\left[s_T(x) \neq s_T(y), \sum_{t=1}^{T-1} s_t(x) = \sum_{t=1}^{T-1} s_t(y)\right],$$

where in (v) we have used equation 11. It remains now to bound

$$\mathbb{P}[s_T(x_1, \ldots, x_T) \neq s_T(y_1, \ldots, y_T), \overline{s}_{T-1}(x_1, \ldots, x_{T-1}) = \overline{s}_{T-1}(y_1, \ldots, y_{T-1})]$$

$$= \sum_{t=1}^{T} \mathbb{P}\left[s_T(x_1, \ldots, x_T) \neq s_T(y_1, \ldots, y_T), \overline{s}_{T-1}(x_1, \ldots, x_{T-1}) = \overline{s}_{T-1}(y_1, \ldots, y_{T-1}) = \frac{t}{T}\right].$$

First, let us define $\overline{x}_T = \frac{1}{T}\sum_{t=1}^{T} x_t$ and $\overline{y}_T = \frac{1}{T}\sum_{t=1}^{T} y_t$. Notice that $\|\overline{x}_T - \overline{y}_T\|^2 \leq \overline{h}_T$. Now, we have

$$\left\{s_T(x_1, \ldots, x_T) \neq s_T(y_1, \ldots, y_T), \overline{s}_{T-1}(x_1, \ldots, x_{T-1}) = \overline{s}_{T-1}(y_1, \ldots, y_{T-1})) = \frac{t}{T}\right\}$$

$$= \left\{\operatorname{sign}\left(w^\top \overline{x}_T - \theta \frac{t}{T}\right) \neq \operatorname{sign}\left(w^\top \overline{y}_T - \theta \frac{t}{T}\right)\right\}$$

$$= \left\{w^\top \overline{x}_T > \theta \frac{t}{T},\, w^\top \overline{y}_T \leq \theta \frac{t}{T}\right\} \cup \left\{w^\top \overline{x}_T \leq \theta \frac{t}{T},\, w^\top \overline{y}_T > \theta \frac{t}{T}\right\}$$

$$= \left\{\sqrt{n} w^\top \frac{\overline{x}_T}{\|\overline{x}_T\|} > \sqrt{n}\theta \frac{t}{T\|\overline{x}_T\|},\, \sqrt{n} w^\top \frac{\overline{y}_T}{\|\overline{y}_T\|} \leq \sqrt{n}\theta \frac{t}{T\|\overline{y}_T\|}\right\} \quad (13)$$

$$\cup \left\{\sqrt{n} w^\top \frac{\overline{x}_T}{\|\overline{x}_T\|} \leq \sqrt{n}\theta \frac{t}{T\|\overline{x}_T\|},\, \sqrt{n} w^\top \frac{\overline{y}_T}{\|\overline{y}_T\|} > \sqrt{n}\theta \frac{t}{T\|\overline{y}_T\|}\right\}.$$

Define $\overline{X} = \sqrt{n} w^\top \frac{\overline{x}_T}{\|\overline{x}_T\|}$ and $\overline{Y} = \sqrt{n} w^\top \frac{\overline{y}_T}{\|\overline{y}_T\|}$ and note that both are standard Gaussians. Their correlation is

$$\rho = \left\langle \frac{\overline{x}_T}{\|\overline{x}_T\|}, \frac{\overline{y}_T}{\|\overline{y}_T\|} \right\rangle$$

$$= \frac{1}{2}\left(\frac{\|\overline{x}_T\|}{\|\overline{y}_T\|} + \frac{\|\overline{y}_T\|}{\|\overline{x}_T\|} - \frac{\|\overline{x}_T - \overline{y}_T\|^2}{\|\overline{x}_T\|\|\overline{y}_T\|}\right) \quad (14)$$

$$\geq \frac{1}{2}\left(\frac{\|\overline{x}_T\|}{\|\overline{y}_T\|} + \frac{\|\overline{y}_T\|}{\|\overline{x}_T\|} - \frac{\overline{h}_T}{\|\overline{x}_T\|\|\overline{y}_T\|}\right)$$

$$\geq \left(1 - \frac{\overline{h}_T}{2\|\overline{x}_T\|\|\overline{y}_T\|}\right).$$

Using equation 13, we get that

$$\mathbb{P}\left[s_T(x_1, \ldots, x_T) \neq s_T(y_1, \ldots, y_T), \overline{s}_{T-1}(x_1, \ldots, x_{T-1}) = \overline{s}_{T-1}(y_1, \ldots, y_{T-1}) = \frac{t}{T}\right] \quad (15)$$

$$\leq \mathbb{P}\left[\overline{X} > \sqrt{n}\theta \frac{t}{T\|\overline{x}_T\|}, \overline{Y} \leq \sqrt{n}\theta \frac{t}{T\|\overline{y}_T\|}\right] + \mathbb{P}\left[\overline{X} \leq \sqrt{n}\theta \frac{t}{T\|\overline{x}_T\|}, \overline{Y} > \sqrt{n}\theta \frac{t}{T\|\overline{y}_T\|}\right].$$

Using Lemma 5, we can distinguish two cases:

1. In the case $\|\overline{y}_T\| = \|\overline{x}_T\| = \mathcal{O}(\frac{\sqrt{n}}{\log n})$, we get

$$\mathbb{P}\left[\overline{X} > \sqrt{n}\theta\frac{t}{T\|\overline{x}_T\|}, \overline{Y} \leq \sqrt{n}\theta\frac{t}{T\|\overline{y}_T\|}\right] \leq \mathbb{P}\left[\overline{X} > \sqrt{n}\theta\frac{t}{T\|\overline{x}_T\|}\right] \qquad (16)$$

$$\overset{(i)}{\leq} e^{-\theta^2 t^2 \frac{n}{T^2\|\overline{x}_T\|^2}} = \mathcal{O}\left(e^{-\frac{\theta^2 t^2}{T^2}\log^2 n}\right)$$

$$\mathbb{P}\left[\overline{X} \leq \sqrt{n}\theta\frac{t}{T\|\overline{x}_T\|}, \overline{Y} > \sqrt{n}\theta\frac{t}{T\|\overline{y}_T\|}\right] \leq \mathbb{P}\left[\overline{Y} > \sqrt{n}\theta\frac{t}{T\|\overline{y}_T\|}\right]$$

$$\overset{(ii)}{\leq} e^{-\theta^2 t^2 \frac{n}{T^2\|\overline{y}_T\|^2}} = \mathcal{O}\left(e^{-\frac{\theta^2 t^2}{T^2}\log^2 n}\right),$$

where in (i) and (ii) we used Gaussian tail bounds. Combining equation 15 and equation 16, it holds

$$\mathbb{P}\left[s_T(x_1,\ldots,x_T) \neq s_T(y_1,\ldots,y_T), \overline{s}_{T-1}(x_1,\ldots,x_{T-1}) = \overline{s}_{T-1}(y_1,\ldots,y_{T-1}) = \frac{t}{T}\right]$$

$$= \mathcal{O}\left(e^{-\frac{\theta^2 t^2}{T^2}\log^2 n}\right).$$

2. In the case $\|\overline{y}_T\| = \|\overline{x}_T\| = \omega(\frac{\sqrt{n}}{\log n})$, combining equation 14 and equation 15, it holds

$$\mathbb{P}\left[s_T(x_1,\ldots,x_T) \neq s_T(y_1,\ldots,y_T), \overline{s}_{T-1}(x_1,\ldots,x_{T-1}) = \overline{s}_{T-1}(y_1,\ldots,y_{T-1}) = \frac{t}{T}\right]$$

$$\leq \mathbb{P}\left[\overline{X} > \sqrt{n}\theta\frac{t}{T\|\overline{x}_T\|}, \overline{Y} \leq \sqrt{n}\theta\frac{t}{T\|\overline{y}_T\|}\right] + \mathbb{P}\left[\overline{X} \leq \sqrt{n}\theta\frac{t}{T\|\overline{x}_T\|}, \overline{Y} > \sqrt{n}\theta\frac{t}{T\|\overline{y}_T\|}\right]$$

$$\overset{(iii)}{\leq} 2\sqrt{1-\rho^2} + 2\frac{\theta t}{T}\frac{\sqrt{n}}{\|\overline{x}_T\|}\frac{1-\rho}{\rho} + 2\frac{\theta t}{T}\sqrt{n}\frac{|\|\overline{y}_T\| - \|\overline{x}_T\||}{\|\overline{x}_T\|\|\overline{y}_T\|\rho}$$

$$\overset{(iv)}{\leq} 2\sqrt{1+\rho}\sqrt{1-\rho} + 2\frac{\theta t}{T}\frac{\sqrt{n}}{\|\overline{x}_T\|}\frac{1-\rho}{\rho} + 4\frac{\theta t}{T}\sqrt{n}\frac{\sqrt{h_T}}{\|\overline{x}_T\|\|\overline{y}_T\|\rho} \qquad (17)$$

$$\leq 2\sqrt{\frac{\overline{h}_T}{\|\overline{x}_T\|\|\overline{y}_T\|}} + 2\theta\frac{\sqrt{n}}{\|\overline{x}_T\|}\frac{\overline{h}_T}{2\|\overline{x}_T\|\|\overline{y}_T\| - \overline{h}_T} + 4\frac{\theta t}{T}\frac{\sqrt{n}}{\|\overline{x}_T\|}\frac{\sqrt{n}\sqrt{\overline{h}_T}}{2\|\overline{x}_T\|\|\overline{y}_T\| - \overline{h}_T}$$

$$\leq C\sqrt{\frac{\overline{h}_T\log^2 n}{n}} + \frac{\theta t}{T}C\frac{\overline{h}_T\log^2 n}{n} + 4\frac{\theta t}{T}C\sqrt{\frac{\overline{h}_T\log^2 n}{n}}$$

$$\leq C\left(1 + 5\frac{\theta t}{T}\right)\sqrt{\frac{\overline{h}_T\log^2 n}{n}}$$

where in (iii) we have used Lemma 3 and in (iv) we applied the inverse triangular inequality.

Combining equation 15, equation 16 and equation 17, we conclude that

$$\mathbb{P}[s_T(x_1,\ldots,x_T) \neq s_T(y_1,\ldots,y_T), \overline{s}_{T-1}(x_1,\ldots,x_{T-1}) = \overline{s}_{T-1}(y_1,\ldots,y_{T-1})]$$

$$= \sum_{t=1}^{T}\mathbb{P}\left[s_T(x_1,\ldots,x_T) \neq s_T(y_1,\ldots,y_T), \overline{s}_{T-1}(x_1,\ldots,x_{T-1}) = \overline{s}_{T-1}(y_1,\ldots,y_{T-1}) = \frac{t}{T}\right]$$

$$\leq \sum_{t=0}^{T}C\left(1 + 5\frac{\theta t}{T}\right)\sqrt{\frac{\overline{h}_T\log^2 n}{n}}$$

$$\leq CT(1 + 5\theta)\sqrt{\frac{\overline{h}_T\log^2 n}{n}}$$

Finally, using equation 12, we conclude that

$$\mathbb{P}[s_T(x) \neq s_T(y)] \leq \sum_{t=1}^{T-1} C(1+5\theta)\, t \sqrt{\frac{\overline{h}_T}{n}} + \mathbb{P}\left[s_T(x) \neq s_T(y), \sum_{t=1}^{T-1} s_t(x) = \sum_{t=1}^{T-1} s_t(y)\right]$$

$$\leq C(1+\theta)\left(T + \sum_{t=1}^{T-1} t\right) \sqrt{\frac{\overline{h}_T \log^2 n}{n}}$$

$$\leq C(1+\theta)T^2 \sqrt{\frac{\overline{h}_T \log^2 n}{n}}.$$

The proof concludes by noticing that $\overline{h}_T = \lfloor \overline{\nu}_T \rfloor$. For static inputs, the same argument applies, but notice that there the use of Lemma 5 is no longer necessary, since $x_1 = x_2 = \ldots = x_t = x$, which implies that $\overline{x}_t = x$ and, since $x \in \{-1,1\}^n$, we have $\|x\| = \sqrt{n}$ (the same is true for $y \in \{-1,1\}^n$).

## D    EXTENSIONS OF THEOREM 1

In this section, we sketch how to extend Theorem 1 to the case of leakage parameter $\beta = 1$, and when sign activation functions of equation 1 are replaced with Heaviside.

### D.1    CASE $\beta \neq 1$

We outline below how the proof of Theorem 1 extends to the general case $\beta \neq 1$. The overall strategy follows the proof above, but worsening the dependence on $T$. We believe this degradation is an artifact of the proof rather than a true limitation of the model. In our main results $T$ is treated as a constant relative to $n$, and under this regime the conclusions remain unchanged.

1. **Base case.** The case $t = 1$ is unchanged, since it does not depend on $\beta$.

2. **Spike condition at time** $T$. For $t > 1$, the neuron spikes at time $T$ for input $x = (x_t)_{t \in [T]}$ if an event of the form

$$w^\top \sum_{t=1}^{T-1} \beta^{T-t} x_t - \frac{\theta}{2} \sum_{t=1}^{T-1} \beta^{T-t-1}(s_{t-1}+1) \geq 0$$

   holds, with an analogous condition for $y$. This changes the structure of the disagreement event $s_T(x) \neq s_T(y)$.

3. **Key partitioning step.** The analogue of equation 12 becomes

$$\mathbb{P}[s_T(x) \neq s_T(y)] = \mathbb{P}\left[s_T(x) \neq s_T(y), \sum_{t=1}^{T-1} \beta^{T-t-1} s_t(x) \neq \sum_{t=1}^{T-1} \beta^{T-t-1} s_t(y)\right]$$

$$+ \mathbb{P}\left[s_T(x) \neq s_T(y), \sum_{t=1}^{T-1} \beta^{T-t-1} s_t(x) = \sum_{t=1}^{T-1} \beta^{T-t-1} s_t(y)\right]. \tag{18}$$

   Unlike the case $\beta = 1$, the term $\sum_{t=1}^{T-1} \beta^{T-t-1} s_t(x)$ can take $2^{T-1}$ distinct values, which is the main source of the suboptimal dependence on $T$.

4. **Averaged inputs.** Similarly, we define

$$\bar{x}_T^\beta := \frac{1}{T} \sum_{t=1}^{T-1} \beta^{T-t} x_t, \qquad \bar{y}_T^\beta := \frac{1}{T} \sum_{t=1}^{T-1} \beta^{T-t} y_t,$$

   and

$$X^\beta := \sqrt{n}\, w^\top \frac{\bar{x}_T^\beta}{\|\bar{x}_T^\beta\|}, \qquad Y^\beta := \sqrt{n}\, w^\top \frac{\bar{y}_T^\beta}{\|\bar{y}_T^\beta\|}.$$

Similar to equation 14, $X^\beta$ and $Y^\beta$ are standard Gaussians with correlation

$$\rho \geq 1 - \frac{\|\bar{x}_T^\beta - \bar{y}_T^\beta\|^2}{\|\bar{x}_T^\beta\|\|\bar{y}_T^\beta\|}.$$

By triangle inequality,

$$\|\bar{x}_T^\beta - \bar{y}_T^\beta\|^2 \leq \left(\sum_{t=1}^T \frac{\beta^{T-t}}{T}\right)^2 \max_{t \in [T]} h_t.$$

5. **Conditioning on spike-history patterns.** For each pattern $a = (a_t)_{t \in [T-1]} \in \{0, 1\}^{T-1}$, set

$$\bar{a}_T^\beta := \frac{1}{T} \sum_{t=1}^{T-1} \beta^{T-t-1} a_t.$$

As in equation 15, we must bound events of the form

$$\mathcal{E} := \left\{ X^\beta \geq \sqrt{n}\theta \frac{\bar{a}_T^\beta}{\|\bar{x}_T^\beta\|}, \ Y^\beta < \sqrt{n}\theta \frac{\bar{a}_T^\beta}{\|\bar{y}_T^\beta\|} \right\},$$

plus the symmetric case.

6. **Two cases, as in the original proof.** We use the same case distinction from Theorem 1, since the conditions for Lemma 5 still hold for $\bar{x}_T^\beta$ and $\bar{y}_T^\beta$.

   - If $\|\bar{x}_T^\beta\| = \|\bar{y}_T^\beta\| = \mathcal{O}(\sqrt{n}/\log n)$, then

   $$\mathbb{P}[\mathcal{E}] \leq e^{-\theta^2 \bar{a}_T^\beta \log^2 n}.$$

   - If $\|\bar{x}_T^\beta\| = \|\bar{y}_T^\beta\| = \omega(\sqrt{n}/\log n)$, the same argument works after replacing $t/T$ with $\bar{a}_T^\beta$ and $\bar{h}_t$ with $\left(\sum_{t=1}^T \frac{\beta^{T-t}}{T}\right)^2 \max_t h_t$.

   Summing over all $2^{T-1}$ patterns gives

   $$\mathbb{P}[s_T(x) \neq s_T(y)] \leq C\, 2^{T-1} \theta \sqrt{\frac{\max_{t \in [T]} h_t \log^2 n}{n}}.$$

7. **Conclusion.** Since our main results assume $T$ is constant relative to $n$, the conclusions remain the same. Improving the dependence on $T$ would require a sharper analogue of equation 15, which appears challenging due to the temporal correlations induced by the shared weights.

## D.2 Heaviside activations

The extension to Heaviside activations requires tracking the norms of the input vectors carefully, but the core argument is unchanged. For clarity, we outline the argument for the simpler case $t = 1$; for $t > 1$, the same reasoning applies exactly as in the proof of Theorem 1.

1. **Setup.** Let $x, y \in \{0, 1\}^n$ with Hamming distance $h = d_H(x, y) = \mathcal{O}(\sqrt{n})$. Define

   $$X = w^\top x, \qquad Y = w^\top y,$$

   and consider the disagreement event

   $$\mathcal{E} := \{X \geq \theta, Y < \theta\} \cup \{X < \theta, Y \geq \theta\}.$$

2. **Correlation structure.** Partition the coordinates as

   $$I = \{i : x_i = y_i = 1\}, \quad J_1 = \{i : x_i = 1, y_i = 0\},$$
   $$J_2 = \{i : x_i = 0, y_i = 1\}, \quad J_3 = \{i : x_i = y_i = 0\}.$$

Then $h = |J_1| + |J_2|$, and

$$X \sim \mathcal{N}\left(0, \frac{|I| + |J_1|}{n}\right), \qquad Y \sim \mathcal{N}\left(0, \frac{|I| + |J_2|}{n}\right).$$

The normalized variables are standard Gaussians with correlation

$$\rho = \frac{|I|}{\sqrt{|I| + |J_1|}\sqrt{|I| + |J_2|}}.$$

Hence, we need to bound

$$p := \mathbb{P}\left[\frac{\sqrt{n}}{\sqrt{|I| + |J_1|}}X \ge \theta\frac{\sqrt{n}}{\sqrt{|I| + |J_1|}}, \ \frac{\sqrt{n}}{\sqrt{|I| + |J_2|}}Y < \theta\frac{\sqrt{n}}{\sqrt{|I| + |J_2|}}\right].$$

3. **Case 1: $|I| = o(n)$.** Since $|J_1| + |J_2| = h = \mathcal{O}(\sqrt{n})$, we have $|J_1|, |J_2| = \mathcal{O}(\sqrt{n})$. Thus,

$$p \le \mathbb{P}[X \ge \theta] \le \exp\left(-\frac{\theta^2}{2}\frac{n}{|I| + |J_1|}\right),$$

which decays exponentially, mirroring Case 1 in the proof of Theorem 1.

4. **Case 2: $|I| = \omega(n)$.** In this regime, $\rho$ is close to 1. Applying Lemma 3 yields

$$p \le \sqrt{1 - \rho^2} + \theta\frac{\sqrt{n}}{|I| + |J_1|}\frac{1 - \rho}{\rho} + \theta\frac{\sqrt{n}\,||J_1| - |J_2||}{\sqrt{|I| + |J_1|}\sqrt{|I| + |J_2|}\,\rho}.$$

Using $|I| = \omega(n)$ and $h = \mathcal{O}(\sqrt{n})$ gives $p = \mathcal{O}\left(\sqrt{\frac{h}{n}}\right)$ which matches the scaling from our original proof. Hence the same stability bound holds for Heaviside activations.

## D.3 PROOF OF THEOREM 2

Let us denote $\overline{x}_T = \frac{1}{T}\sum_{k=1}^{T} x_k$, $\overline{y}_T = \frac{1}{T}\sum_{k=1}^{T} y_k$ and $d_H(x_t, y_t) = \lfloor \nu_t n \rfloor$ with $\nu_t \in [0, 1]$. Let us define $\nu := \max_{t \in [T]} \nu_t$ and assume $\nu = \mathcal{O}(\frac{1}{\sqrt{n}})$. We follow a similar strategy to that of Jonasson et al. (2023). Notice that, for all $l \in [L]$, the probability that $s_{T,i}^{(l)}(x_1, \ldots, x_T, W)$ and $s_{T,i}^{(l)}(y_1, \ldots, y_T, W)$ differs depends only on the number of neurons that have at least one disagreement at any time at layer $l - 1$ and not on where they disagree. We define $D_T^{(l)}$ as the number of neurons at layer $l$ that have at least one disagreement at any time, that is $D_T^{(l)} := \frac{1}{2}\sum_{i=1}^{n} \max_{k \in [T]} |s_{k,i}^{(l)}(x_1, \ldots, x_k, W) - s_{k,i}^{(l)}(y_1, \ldots, y_k, W)|$. With this, $D_T^{(l)}$ is a Markov chain with $n + 1$ states, where $D_T^{(l)} = 0$ is an absorbing state. More precisely, let us denote with $M_i := \frac{1}{2}\max_{k \in [T]} |s_{k,i}^{(l)}(x_1, \ldots, x_k, W) - s_{k,i}^{(l)}(y_1, \ldots, y_k, W)|$. Notice that $M_1, \ldots, M_n$ are independent random variables with value in $\{0, 1\}$. In particular, exploiting Theorem 1, it holds that

$$\mathbb{P}[M_i = 1] = \mathbb{P}\left[\max_{k \in [T]} |s_{k,i}^{(l)}(x_1, \ldots, x_k, W) - s_{k,i}^{(l)}(y_1, \ldots, y_k, W)| = 2\right]$$

$$\le T^3 C(1 + \theta)\sqrt{\frac{\overline{h}_T^{(l-1)} \log^2 n}{n}}$$

$$\le T^3 C(1 + \theta)\sqrt{\frac{D_T^{(l-1)} \log^2 n}{n}}$$

for all $i \in [n]$. Therefore, $D_T^{(l)} | D_T^{(l-1)} = k$ is a Binomial random variable upper bounded in the stochastic sense by

$$\tilde{D}_T^{(l)} | D_T^{(l-1)} = k \sim \text{Bin}\left(n, T^3 C(1 + \theta)\sqrt{\frac{k \log^2 n}{n}}\right), \tag{19}$$

thanks to Lemma 2. We invite the reader to see Definition 5 for a rigorous definition of *smaller in the usual stochastic order*. We write now $h_t = \lfloor \nu_t n \rfloor$ for some $\nu_t \in [0,1]$ and define $\nu = \max_{k \in [T]} \nu_t$ the maximum number of input disagreements.

We proceed by induction over the depth dimension $l \in [L]$.

- Let us start from the base case $l = 1$. We want to estimate

$$\mathbb{P}_W \left( D_T^{(1)} \geq \nu^{1/4} n \log n \right) = \sum_{k=1}^{n} \mathbb{P}_W \left( D_T^{(1)} \geq \nu^{1/4} n \log n \mid D_T^{(0)} = k \right) \mathbb{P}_W \left( D_T^{(0)} = k \right)$$

$$\leq \sum_{k=1}^{\lfloor \nu n \rfloor} \mathbb{P}_W \left( D_T^{(1)} \geq \nu^{1/4} n \log n \mid D_T^{(0)} = k \right) \mathbb{P}_W \left( D_T^{(0)} = k \right) + \mathbb{P}_W \left( D_T^{(0)} > \lfloor \nu n \rfloor \right)$$

$$\overset{(i)}{\leq} \mathbb{P}_W \left( \tilde{D}_T^{(1)} \geq \nu^{1/4} n \log n \mid D_T^{(0)} = \lfloor \nu n \rfloor \right) ,$$

where in (i) we used the stochastic dominance and the monotonicity of the probability appearing in equation 19. Now, combining Theorem 3 with $\varepsilon = 1$, which is admissible given that $\frac{\nu^{-1/4}}{CT^3(1+\theta)} > 2$ and $\nu \leq \frac{1}{\sqrt{n}}$ for $n$ large enough, we conclude

$$\mathbb{P}_W \left( \tilde{D}_T^{(1)}(x,y) \geq \nu^{1/4} n \log n \mid D_T^{(0)}(x,y) = \lfloor \nu n \rfloor \right) \leq e^{-\frac{1}{3}\sqrt{\nu}n} ,$$

- Let us now assume by induction that

$$\mathbb{P}_W \left( D_T^{(l)} \geq \nu^{\frac{1}{2^{2l}}} n \log n \right) \leq l e^{-\frac{1}{3} \nu^{\frac{1}{2^{2l-1}}} n} . \tag{20}$$

Then, we have

$$\mathbb{P}_W \left( D_T^{(l+1)} \geq \nu^{1/2^{2(l+1)}} n \log n \right)$$

$$= \sum_{k=1}^{n} \mathbb{P}_W \left( D_T^{(l+1)} \geq \nu^{1/2^{2(l+1)}} n \log n \mid D_T^{(l)} = k \right) \mathbb{P}_W \left( D_T^{(l)} = k \right)$$

$$\leq \sum_{k=1}^{\left\lfloor \nu^{\frac{1}{2^{2l}}} n \log n \right\rfloor} \mathbb{P}_W \left( D_T^{(l+1)} \geq \nu^{1/2^{2(l+1)}} n \log n \mid D_T^{(l)} = k \right) \mathbb{P}_W \left( D_T^{(l)} = k \right) +$$

$$+ \mathbb{P}_W \left( D_T^{(l)} > \left\lfloor \nu^{\frac{1}{2^{2l}}} n \log n \right\rfloor \right) \tag{21}$$

$$\leq \sum_{k=1}^{\left\lfloor \nu^{\frac{1}{2^{2l}}} n \log n \right\rfloor} \mathbb{P}_W \left( \tilde{D}_T^{(l+1)} \geq \nu^{\frac{1}{2^{2(l+1)}}} n \log n \mid D_T^{(l)} = k \right) \mathbb{P}_W \left( D_T^{(l)} = k \right) + l e^{-\frac{1}{3} \nu^{\frac{1}{2^{2l-1}}} n}$$

$$\leq \mathbb{P}_W \left( \tilde{D}_T^{(l+1)} \geq \nu^{\frac{1}{2^{2(l+1)}}} n \log n \mid D_T^{(l)} = \left\lfloor \nu^{\frac{1}{2^{2l}}} n \log n \right\rfloor \right) + l e^{-\frac{1}{3} \nu^{\frac{1}{2^{2l-1}}} n} .$$

Now, using Theorem 3 with $\varepsilon = 1$, which is admissible because $\frac{\nu^{-\frac{1}{2^{2(l+1)}}}}{\sqrt{\log n} T^3 C(1+\theta)} > 2$, and $\nu \leq \frac{1}{\sqrt{n}}$, for $n$ large enough, we conclude that

$$\mathbb{P}_W \left( \tilde{D}_T^{(l+1)}(x,y) \geq \nu^{\frac{1}{2^{2(l+1)}}} n \log n \mid D_T^{(l-1)}(x,y) = \left\lfloor \nu^{\frac{1}{2^{2l}}} n \log n \right\rfloor \right) \leq l e^{-c\nu^{\frac{1}{2^{2l+1}}} n} . \tag{22}$$

Hence, combining equation 21 and equation 22, we conclude that

$$\mathbb{P}_W \left( \tilde{D}_T^{(l)}(x,y) \geq \nu^{\frac{1}{2^{2(l+1)}}} n \log n \right) \leq l e^{-c\nu^{\frac{1}{2^{2l+1}}} n} .$$

Now, if $L = 1$, we can apply directly Theorem 1 obtaining that

$$\mathbb{P}_W \left( f^{1,T}(x, W) \neq f^{1,T}(y, W) \right)$$

$$= \mathbb{P}_W \left( \arg\max \sum_{t=1}^T s_{t,i}^{(1)} \left( (x_t)_{t \in [T]}, W \right) \neq \arg\max \sum_{t=1}^T s_{t,i}^{(1)} \left( (y_t)_{t \in [T]}, W \right) \right)$$

$$\leq \mathbb{P}_W \left( \max_{k \in [T], i \in [n_L]} |s_{k,i}^{(1)} \left( (x_t)_{t \in [T]}, W \right) - s_{k,i}^{(1)} \left( (y_t)_{t \in [T]}, W \right)| > 0 \right)$$

$$\leq n_L T^3 C (1 + \theta) \log n \sqrt{\nu}.$$

If $L \geq 2$, using Theorem 1, we conclude that

$$\mathbb{P}_W \left( f^{L,T}(x, W) \neq f^{L,T}(y, W) \right)$$

$$\leq \mathbb{P}_W \left( \max_{k \in [T], i \in [n_L]} |s_{k,i}^{(L)} \left( (x_t)_{t \in [T]}, W \right) - s_{k,i}^{(L)} \left( (y_t)_{t \in [T]}, W \right)| > 0 \right)$$

$$\leq \mathbb{P}_W \left( \max_{k \in [T], i \in [n_L]} |s_{k,i}^{(L)} \left( (x_t)_{t \in [T]}, W \right) - s_{k,i}^{(L)} \left( (y_t)_{t \in [T]}, W \right)| \, \Big| \, D^{(L-1)} \leq \lfloor \nu^{\frac{1}{2^{2L}}} n \log n \rfloor \right)$$

$$+ (L - 1) e^{-c \nu^{\frac{1}{2^{2L-1}}} n}$$

$$\leq n_L T^4 C (1 + \theta) \nu^{\frac{1}{2^{2L+1}}} \log^{3/2} n + (L - 1) e^{-c \nu^{\frac{1}{2^{2L-1}}} n},$$

which concludes the proof.

### D.4 Proof of Corollary 1

Let $x, y \in \{-1, 1\}^n$ be static inputs. Define $N(\xi) = \frac{1}{2} \sum_{i=1}^n (\xi_i + 1)$. Then, we have for $x \sim \text{Unif}(\{-1, 1\}^n)$ and $\xi \sim \text{Rad}(1 - \nu)$

$$\mathbf{ENS}_\nu \left( \{ f^{L,T}(\cdot, W) \}_{W \sim \mathcal{N}(0, I_d)} \right)$$

$$= \mathbb{E}_{x,\xi} \left[ \mathbb{P}_W \left( f^{L,T}(x, W) \neq f^{L,T}(x \odot \xi, W) \right) \right]$$

$$= \mathbb{E}_x \left[ \mathbb{P}_W \left( f^{L,T}(x, W) \neq f^{L,T}(x \odot \xi, W) \Big| N(\xi) \leq \sqrt{n} \right) \mathbb{P}_\xi (N(\xi) \leq \sqrt{n}) \right]$$

$$+ \mathbb{E}_x \left[ \mathbb{P}_W \left( f^{L,T}(x, W) \neq f^{L,T}(x \odot \xi, W) \Big| N(\xi) > \sqrt{n} \right) \mathbb{P}_\xi (N(\xi) > \sqrt{n}) \right]$$

$$\leq C_{T,\theta} \nu'^{\frac{1}{2^{2L+1}}} \log^{3/2} n + (L - 1) e^{-c \nu'^{\frac{1}{2^{2L-1}}} n} + e^{-\frac{1}{4} \sqrt{n}}.$$

In the last line we used Theorem 2 with $\nu = \nu'$, which applies because $N(\xi) \leq \lfloor \sqrt{n} \rfloor$ is equivalent with $d_H(x, x \odot \xi) = \lfloor \sqrt{n} \rfloor$, and $\nu' \leq \frac{1}{\sqrt{n}}$ (by assumption). We also used that

$$\mathbb{P}_\xi (N(\xi) > \lfloor \sqrt{n} \rfloor) \leq e^{-\frac{1}{4} \sqrt{n}},$$

which follows from applying the Chernoff bound (in Theorem 3) to the random variable $N(\xi) \sim \text{Bin}(n, \nu')$. The statement about the expected degree concentration follows applying Lemma 6, using the bound on the expected noise sensitivity above.

## E  Additional Experiments

In this appendix, we report additional experiments that complement the results presented in the main text. These include further evaluations of noise sensitivity and analyses of (s)IF spiking neural networks under different settings.

**Single (s)LIF Neuron.**  We present additional experiments on (s)LIF spiking neuron. These results complement the main text by providing additional empirical support for the claims made in the experiment section. In Figure 7, we report results for the neuron with threshold $\theta = 0$. In this case, we observe that the neuron either fires at all times or does not fire at all. This is consistent with the experiments, as the sensitivity remains constant over time.

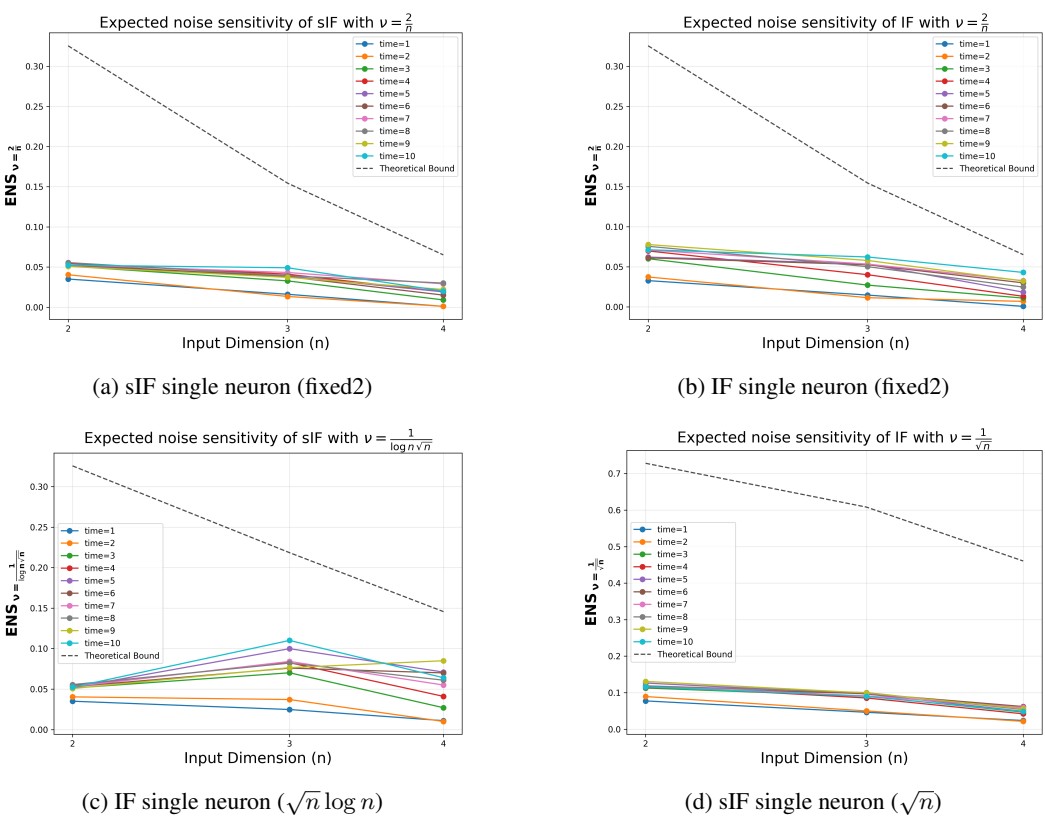

Figure 5: Noise sensitivity $\mathbf{ENS}_{2/n}$ and $\mathbf{ENS}_{1/(\sqrt{n}\log n)}$ for different input dimensions $n$ for sIF and IF neurons with $\theta = 0.5$ and $T = 10$. **(a)** $\mathbf{ENS}_{2/n}$ for sIF neuron. **(b)** $\mathbf{ENS}_{2/n}$ for IF neuron; **(c)** $\mathbf{ENS}_{1/(\sqrt{n}\log n)}$ for sIF neuron. **(d)** $\mathbf{ENS}_{1/(\sqrt{n}\log n)}$ for IF neuron.

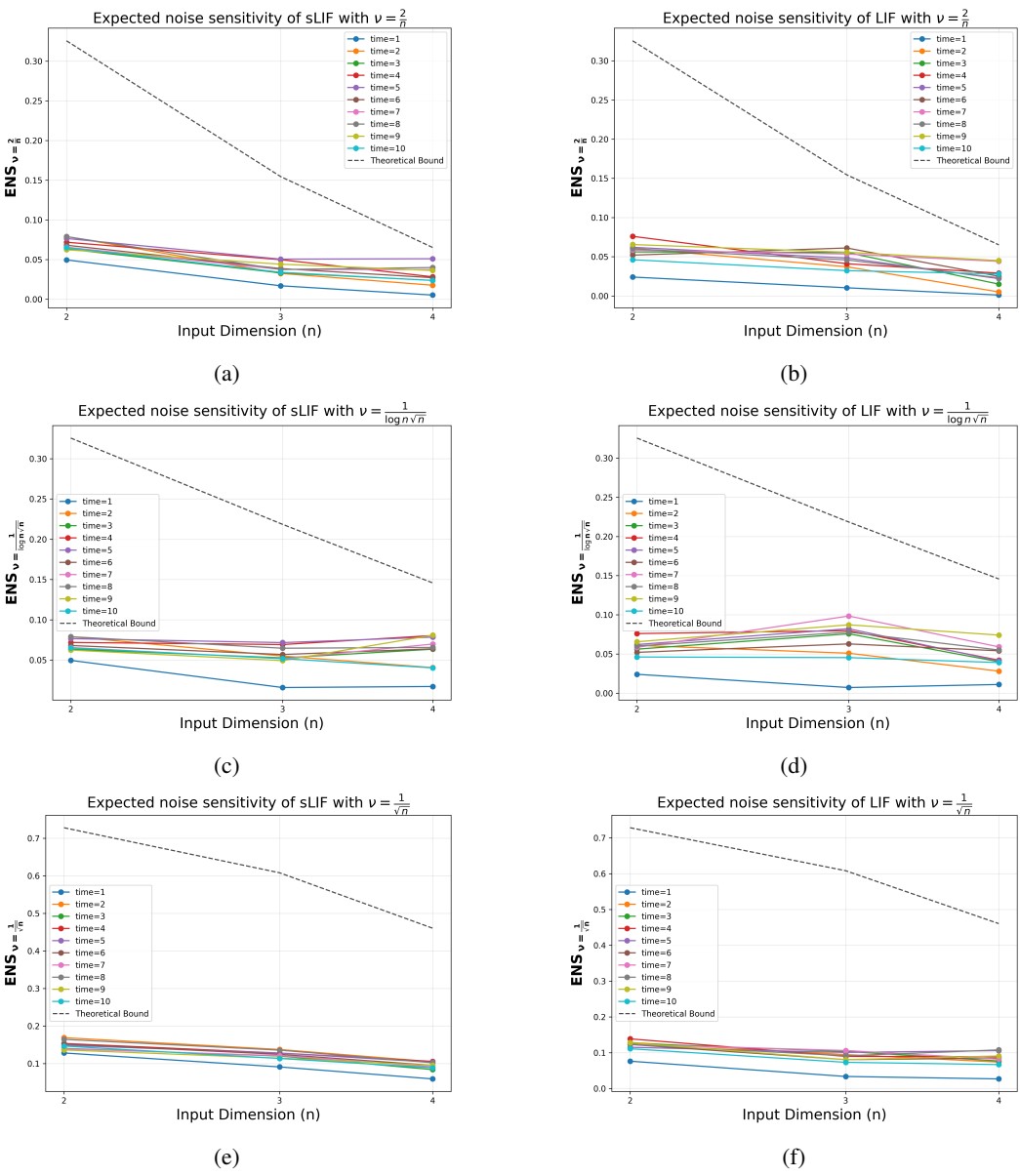

Figure 6: Noise sensitivity $\mathbf{ENS}_{2/n}$ and $\mathbf{ENS}_{1/(\sqrt{n}\log n)}$ for different input dimensions $n$ for sLIF and LIF neurons with $\theta = 0.5, T = 10$ and $\beta = 0.5$. **(a)** $\mathbf{ENS}_{2/n}$ for sLIF neuron. **(b)** $\mathbf{ENS}_{2/n}$ for LIF neuron; **(c)** $\mathbf{ENS}_{1/(\sqrt{n}\log n)}$ for sLIF neuron. **(d)** $\mathbf{ENS}_{1/(\sqrt{n}\log n)}$ for LIF neuron.

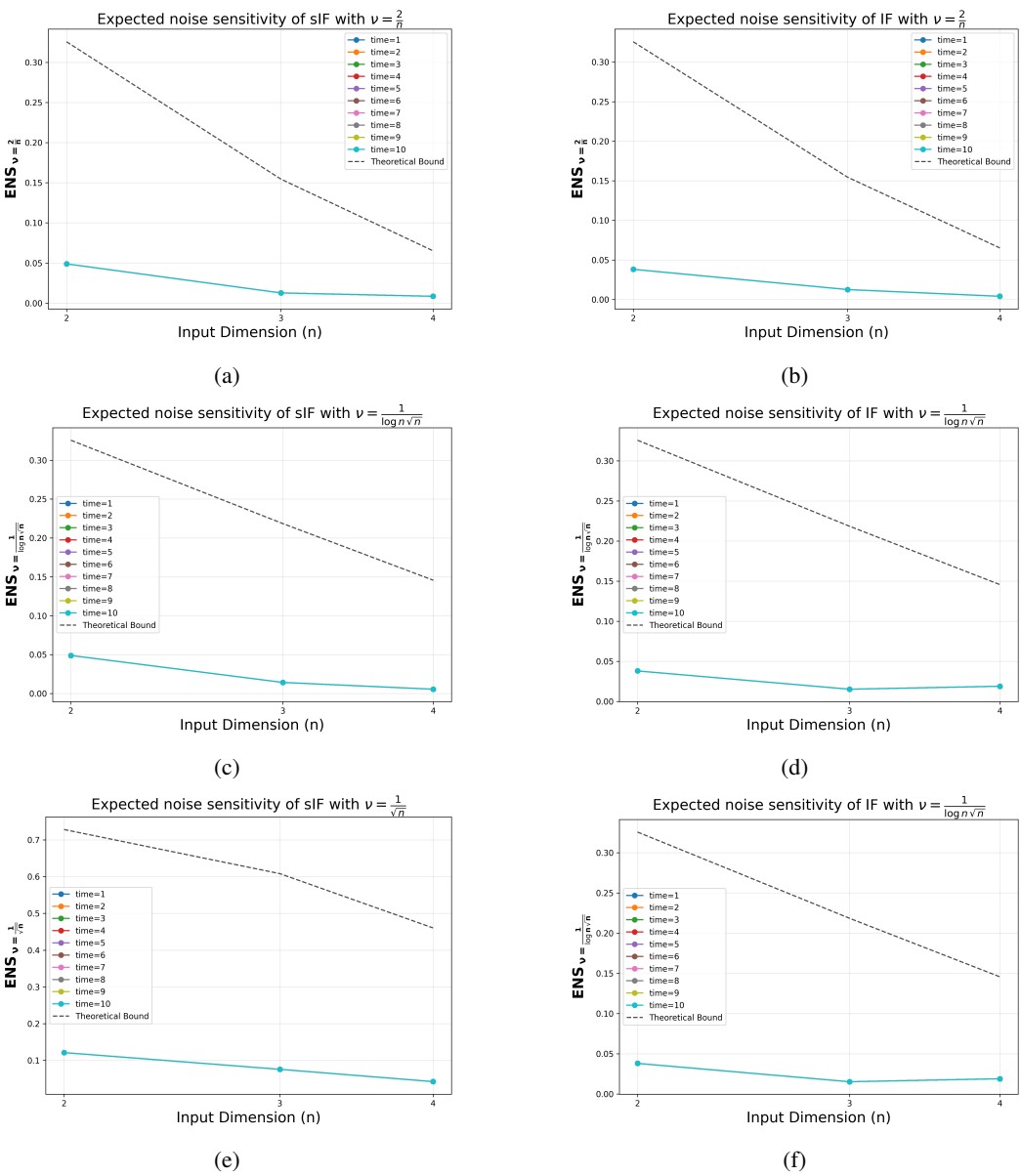

Figure 7: Noise sensitivity $\mathbf{ENS}_{2/n}$ and $\mathbf{ENS}_{1/(\sqrt{n}\log n)}$ for different input dimensions $n$ for sIF and IF neurons with $\theta = 0$ and $T = 10$. **(a)** $\mathbf{ENS}_{2/n}$ for sIF neuron. **(b)** $\mathbf{ENS}_{2/n}$ for IF neuron; **(c)** $\mathbf{ENS}_{1/(\sqrt{n}\log n)}$ for sIF neuron. **(d)** $\mathbf{ENS}_{1/(\sqrt{n}\log n)}$ for IF neuron; **(e)** $\mathbf{ENS}_{1/(\sqrt{n})}$ for sIF neuron. **(d)** $\mathbf{ENS}_{1/(\sqrt{n})}$ for IF neuron.

**Deep (s)IF SNN.** We present additional experiments on deep (s)IF spiking neural networks. These results complement the main text by extending the analysis of noise sensitivity to multi-layer architectures.

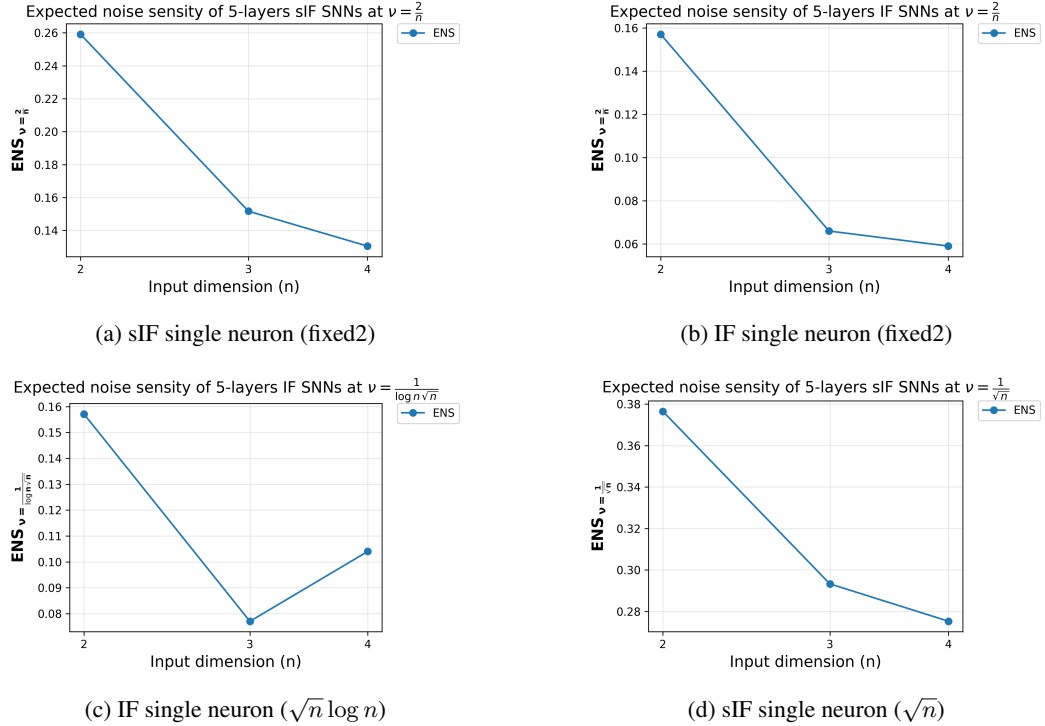

(a) sIF single neuron (fixed2)

(b) IF single neuron (fixed2)

(c) IF single neuron $(\sqrt{n}\log n)$

(d) sIF single neuron $(\sqrt{n})$

Figure 8: Noise sensitivity $\mathbf{ENS}_{1/\sqrt{n}}$ for different input dimensions $n$ for 5-layers sIF and IF neural networks with $\theta = 0.5$ and $T = 10$. **(a)** $\mathbf{ENS}_{2/n}$ for 5-layers sIF neural network (log-scale x-axis); **(b)** $\mathbf{ENS}_{1/(\sqrt{n}\log n)}$ for 5-layers sIF neural network (log-scale x-axis).

