# OpenReview forum: "Random Spiking Neural Networks are Stable and Spectrally Simple"
_ICLR.cc/2026/Conference — ICLR 2026 Poster_

### Official Review · Reviewer_KsVi · 2025-10-27

**Soundness:** 3
**Presentation:** 3
**Contribution:** 3
**Rating:** 6
**Confidence:** 3

**Summary:**

The paper builds on previous theoretical work and investigates the stability of random Leaky integrate-and-fire spiking neural network classifiers. The question is, how is the output of the network sensitive to the changes in input and when the changes in the input change the classification. While the majority of the work is theoretical, a minor part consists of numerical experiments that test the validity of theoretically computed bounds on training hyperparameters. It is found that random spiking networks are stable, in the sense that small changes to the inputs do not change the classification outcomes.

**Strengths:**

While the paper builds on previous results, it also brings original contribution to previous works. The paper is logically constructed, motivates well the interest of the topic and brings significant results on the stability of SNN classifiers. The paper is rigorous and brings a valid theoretical advancement to the understanding of spring neural networks. Rigour and clarity are, among others, achieved through defining and explaining all quantities used in equations.

**Weaknesses:**

A sizeable portion of the paper presents previous results. This is in itself not a problem, as it serves the purpose of presenting a complete theory and because new results would be hard to put into context otherwise. However, on several occasions it remains to some degree unclear what is a previous result  and what is a new contribution. Authors could be more specific about this as they unfold their results.
Moreover, some results remain unclear and could be better explained. In some places, the presentation would profit from clarification, better intuitive explanations and on further comments of obtained results.  More commentary and intuitive explanation could be provided along the Results part of the paper to allow to the reader to better understand and appreciate the significance of the results. Finally, the paper does not have a significant limitation section, which would be important to have.

Minor:
The y-axis of Figures 1 and 2 is "ENS" and the y-axis of Figure 3 is "noise sensitivity".
x-axis of Figures 1 and 2 report the number of features, but in the text they are referred to as the "input dimensions".  It is advisable to unify the axis notation.

**Questions:**

1) [line 162] Authors give a definition of what they call a recurrent network, but to me it seems that they define a multilayer feedforward spiking network - a network where each neuron in layer $l$ receives spiking output from neurons in the previous layer $l-1$. In what sense is this a recurrent network?

2) Proof sketch : step 3 reads somehow unclear, can the authors explain better what is done in the last step?

3) Why does Figure 2 not show the bounds?

4) Figure 3 shows the sensitivity as a function of \nu. Why is that?

5) The dependance of the result on log n in Theorem 1 is not clear to me. Can authors explain?

6) Authors perform their analysis on random spiking networks. Would results apply to spiking networks with structured connectivity (e.g. Koren and Panzeri, NeurIPS 2022; Urdu et al., ICANN 2025)? If not, what is fundamentally different in structured vs random networks that would prevent such generalisation of your results to structured networks? If authors find the question relevant, I suggest discussing it in the revision.

---

> ### Author Response · Authors · 2025-11-21
>
> We thank the reviewer for their time and the useful feedback. We address each question and weakness individually.
>
>  **Comments on the weaknesses**
>
> 1.  This work exploits concepts borrowed from different fields,  from computational neuroscience to mathematics. To ensure that both the communities have the necessary background and vocabulary to understand this contribution, we believe that it essential to formally introduced both the LIF model and the concepts of sensitivity analysis and Fourier analysis of Boolean functions. Even in the context of Boolean analysis the definitions of expected noise sensitivity and expected spectral concentration are not standard (the usual setting is for a fixed function), as far as we know. All the presented results Theorem 1, Theorem 2 and Corollary 1 are new, and they constitute the core of this work. Additionally, some of the presented lemmas could be used to analyze other properties of this model. We will add a limitation section in the Conclusions in the final version after the discussion phase.
>
> 2. We followed the reviewer's suggestions on the experiments, and we have already updated them in the revised version.
>
> **Answer to the questions**
>
> 1. LIF spiking neural networks are often considered recurrent neural networks because of the time dimension and how the neuron dynamics evolve. Indeed, the membrane potential of a single spiking neuron acts as a hidden state storing information from the past. This is exactly what recurrent hidden units do, but following a different dynamic, see Figure 6. pag 1022 in
>
>     "J. K. Eshraghian et al., Training Spiking Neural Networks Using Lessons From Deep Learning," in Proceedings of the IEEE, vol.     111, no. 9, pp. 1016-1054, Sept. 2023".
>
> 2. The last inequality in Step 3 in the proof of Theorem 1 follows from
>
>       $$ \mathbb{P}[sign(w^\top x - \theta)\neq sign(w^\top y - \theta)] = \mathbb{P}[X>\theta, Y\leq \theta] + \mathbb{P}[X\le \theta, Y> \theta]\le \Phi_2 \big(-\theta,\theta;2\nu_1-1\big) + \Phi_2(\theta,-\theta;2\nu_1-1)$$
>
>     Since $|x|=|y|$ the lower bound follows from Lemma 3 as in equation (10). We pointed out Lemma 3 and equation (10) in the sketch of the proof in this revised version of the paper.
>
> 3. Figure 2 does not show the theoretical bound since it is way above the empirical stability bound already with L=5. This reflect directly the limitation addressed in the Remark at line 363.
>
> 4. In Figure 3, $\nu$ refers to the fraction of components that are (in mean) flipped in the input. The notation is consistent with Theorem 2 and is specified in lines 434-435 in the current version.
>
> 5. Theorem 1 directly follows from the first case in the proof on page 17, in particular, equation 8. Indeed, we notice that it is necessary to split the proof into two distinguished cases which exploit two different approaches based on the values of $|\bar{x}_T|$. The first approach strongly rely on a tail bound to control the events $\lbrace\bar{X} > \sqrt{n} \theta \frac{t}{T|\bar{x}_T|}, \bar{Y} \le  \sqrt{n} \theta \frac{t}{T|\bar{y}_T|}\rbrace$ and  $\lbrace\bar{X} \le \sqrt{n} \theta \frac{t}{T|\bar{x}_T|}, \bar{Y} >  \sqrt{n} \theta \frac{t}{T|\bar{y}_T|}\rbrace$. However, when $|\bar{x}_T| = |\bar{y}_T|= \Omega(\sqrt{n})$, this approach is not informative since these two events will have a non-vanishing (in $n$) probability. Therefore, in that case this case it is necessary to bound these probabilities following another approach (case 2.) based on Lemma 3. A standard way to distinguish between "of order $\sqrt{n}$" or not is to divide  $\sqrt{n}$ by a $\log n$ factor, obtaining the distinctions between $|\bar{x}_T| = |\bar{y}_T|= \mathcal{O}(\sqrt{n}/\log n)$ and $|\bar{x}_T| = |\bar{y}_T|= \omega(\sqrt{n}/\log n)$ which directly produces the dependence of $\log n$ in Theorem 1.
>
> 6. We believe that our approach can be generalized to other structured connectivity as long as no extra dependencies are introduced. For instance, this does not allow convolutions or skip connections. Regarding the "Koren, Veronika, and Stefano Panzeri. "Biologically plausible solutions for spiking networks with efficient coding." Advances in neural information processing systems 35 (2022): 20607-20620." that the reviewer mentions, we couldn't find a specific connectivity structure, and it seems more connected with a different loss function for different neuron types. Regarding the other reference, we couldn't locate the specific reference in the book of ICANN 2025 cited by the reviewer.

---

> > ### Comment · Reviewer_KsVi · 2025-11-28
> > **additional clarifications needed**
> >
> > Thank you for your reply.
> >
> > My questions have been answered, besides the last question. Regarding the point 6., in recurrent spiking networks with structured connectivity, as for example in references [1-3], the connectivity is structured in relation to the pairwise similarity of the tuning of neurons $i$ and $j$ to the stimulus. In the simplest case, $J_{ij} = \vec{w}_i^T \vec{w}_j$, where $\vec{w}_i$ is the vector of tuning to the stimulus features and $T$ is the transpose. This connectivity structure s derived analytically. In other numerous cases, the network's connectivity is numerically trained to optimise the performance of the network on a specific task or a set of tasks, and the resulting connectivity will not be random. Does connectivity structure impose a caveat to your analysis? If yes, why and where can we see that? If not, also why and where can we see that?
> >
> > [1] Bourdoukan, R., & Deneve, S. (2015). Enforcing balance allows local supervised learning in spiking recurrent networks. Advances in Neural Information Processing Systems, 28.
> > [2] Kadmon, J., Timcheck, J., & Ganguli, S. (2020). Predictive coding in balanced neural networks with noise, chaos and delays. Advances in neural information processing systems, 33, 16677-16688.
> > [3] Koren, V., & Panzeri, S. (2022). Biologically plausible solutions for spiking networks with efficient coding. Advances in neural information processing systems, 35, 20607-20620.

---

> > > ### Author Response · Authors · 2025-12-02
> > >
> > > We are glad our earlier response was helpful, and address the reviewer’s follow-up questions below.
> > >
> > > In general, connectivity induces strong dependences among the randomly initialized weights, and this complicates a possible extension of our approach. We provide an example showing where the problem lies. We consider a discretization of the model introduced in equations (1) and (2) in [2], with the simplifying assumption that the interaction term $J_{ij}=\frac{w_iw_j}{N}$. In particular, the model is the discrete state space model given by $ \tau h_i(t+1) =  \beta h_i(t) + \frac{1}{N}\sum_{j\ne i} w_i w_j r_j(t-d) + w_i x(t) + \sigma \xi_i(t)$ and $r_i(t)=\Phi(h_i(t))$. Notice that the model in [2] processes scalar sequences, contrary to our setting. Indeed, in the scalar $n=1$ case, correlations between sequences $x(t)$ and $y(t)$ are fixed, there is no possible asymptotic analysis, and a trivial case distinction should work. As a thought experiment, we extend the model to the case where $x(t)$ is a vector, i.e.,
> > >
> > > $$ \tau h_i(t+1) =  \beta h_i(t) + \frac{1}{N}\sum_{j\ne i} w^\top_i w_j r_j(t-d) + w^\top_i x(t) + \sigma \xi_i(t)$$
> > >
> > >  and $r_i(t)=\Phi(h_i(t))$, assuming for simplicity that $\Phi$ is a sign/Heaviside function and $\beta=1, \sigma=0$ and $d=1$. At $t=1$, with the initial condition $h_i(0)=0$, the equation reduces to $r_i(1) = H(w^\top_i x(0))$ and this is analogous to our case at $t=1$. Then, for $t=2$, we have $r_i(2) = H(w^\top_i(\frac{1}{N}\sum_{j\ne i}w_jr_j(1) + x(0)+ x(1)))$. Now, we notice that $r_j(1)$ and $w_j$ are correlated; therefore $(\frac{1}{N}\sum_{j\ne i}w_jr_j(1) + x(0)+ x(1))$ is not Gaussian, and, hence, our results does not directly extend. Moreover, it is not clear what is its distribution. An alternative approach would be to uniformly bound $\frac{1}{N}\sum_{j\ne i}w_jr_j(1)$, different techniques.
> > > We are grateful for the reviewer’s careful consideration of our work and welcome further discussion on this or any additional points.

---

### Official Review · Reviewer_7NY3 · 2025-10-31

**Soundness:** 3
**Presentation:** 3
**Contribution:** 3
**Rating:** 8
**Confidence:** 3

**Summary:**

The paper studies stability of discrete-time leaky integrate-and-fire (LIF) spiking neural networks (SNNs) through Boolean function analysis. It proves that wide, randomly initialized LIF-SNN classifiers are noise-stable on average: small random input bit flips are unlikely to change the predicted class. From this, the authors introduce spectral simplicity—a notion saying most Fourier–Walsh mass concentrates on low degrees—and show random LIF-SNNs are biased toward such “simple” functions. Experiments on single neurons, deep networks, and trained models (MNIST/CIFAR-10) suggest (i) both IF and “signed” IF variants are noise-stable, (ii) depth increases sensitivity but less than the theory’s worst-case bounds, and (iii) training further reduces sensitivity.

**Strengths:**

Stability/robustness theory for spiking models is far less developed than for ANNs; bringing Boolean analysis + Fourier tools here is novel and interesting.

Spectral simplicity offers a principled bridge from stability to a simplicity notion compatible with SNNs and complements prior simplicity-bias results in ANNs.

The signed LIF formulation and spike-count readout are standard enough to be meaningful while still analyzable; proofs acknowledge the reset-by-subtraction complication. Also, if random LIF-SNNs are spectrally simple, that has consequences for generalization priors and for neuromorphic design choices (thresholds, depth, latency).

**Weaknesses:**

The strongest theory is for random weights and static encodings; dynamic inputs are acknowledged but technically harder (partial sums leave the hypercube). This narrows practical reach (e.g., event streams).

The corollary infers low-degree concentration from ENS bounds, but there’s no empirical Fourier probe of trained models to confirm the spectrum actually concentrates as predicted.

**Questions:**

Can parts of the analysis (Theorem 1/2) be ported to the classical Heaviside LIF (without sign) to close the theory–practice gap you highlight empirically? Which steps fail, and can they be patched (e.g., via comparison lemmas)?

Your proofs suggest stability worsens with L and T, yet experiments look kinder. Can you isolate which proof steps (union bounds, reset dependencies) are driving pessimism and provide refined bounds (e.g., martingale or coupling arguments) that track empirical trends?

How sensitive are your results to input distributions? Could the corollary be re-stated under a product measure with bounded correlations, or for common dataset distributions after whitening? You mention generalization is “straightforward”—can you sketch it?

---

> ### Author Response · Authors · 2025-11-21
>
> We thank the reviewer for their time and the useful feedback. We address each question and weaknesses individually.
>
> **Comments on weaknesses**
>
> 1. We want to clarify that Theorem 1, which bounds the sensitive of one single sIF neuron, gives better results for static encoding, but only up to logarithmic terms, which for large $n$ are negligible. On the other hand, Theorem 1 works for dynamic inputs, even if the partial sums leave the hypercube. Lemma 5 takes care of that. We needed this for Theorem 2, since the output after the first layer is always dynamic.
>
> 2. This is an interesting point. However, empirically compute the Fourier-Walsh coefficients in high-dimension is a non-trivial numerical problem. We repeat here our answer to Question 2 of Reviewer 77v2.  We would like to emphasize that estimating Fourier--Walsh coefficients in high dimensions is intrinsically challenging. For a classifier $f : \lbrace-1,1\rbrace^n \to \lbrace-1,1\rbrace$, the coefficients are defined as
>  $$\hat{f}(S) = \mathbb{E}_{x \sim \mathrm{Unif}(\{-1,1\}^n)}[ f(x)\chi_S(x)]$$
>
>     and $ \chi_S(x)=\prod_{i\in S} x_i$.
>     Accurately approximating these expectations via Monte Carlo requires an exponentially large number of samples, analogous to high-dimensional numerical integration. More refined methods, such as the Kushilevitz-Mansour (KM) algorithm, allow identification of *large* coefficients, but they do not scale favorably with the ambient dimension. We are currently trying to obtain reliable estimates of the coefficients, but obtaining good approximations seems to be a research problem in itself. We will appreciate any insights, references, or suggestions that we may have overlooked.
>
> **Answer to the questions**
>
> 1. We agree that this is a point we should further discuss in the final version. The results can be extended, with minimal changes, to the case of Heaviside LIF neurons. Essentially, it requires tracking the norms of the input vectors carefully, distinguishing two cases: when the input norms are $o(n)$ or $\omega(n)$. To avoid repetitions, and for the sake of readability, we invite the reviewer to see our detailed answer to the Question 2 of Reviewer rx3u.  We plan to include that answer as a formal remark with all the details after the discussion phase.
>
> 2. We briefly discuss some of the steps in the proof that worsen the bounds and potential ways to improve them.
> -  *In Theorem 1*: the looser bound is in eq (15). In that bound, a better strategy will be estimate the joint probability by conditioning. However, given the dependencies of the variables involved, it is not clear how to tighten the bound. This introduce a sub-optimal dependence on $T$.
> - *In Theorem 2*: the Markov-Chain $D^{(l)}_T $ used in the proof contains a maximum over time, which could be pessimistic. Additionally, our application of the Chernoff bound requires that $D^{(l)}_T$ is larger than the mean of the Binomial $D^{(l)}_T | D^{(l-1)}_T $. An upper bound on this mean is provided by the inductions step. To improve this one should use either a different Markov chain, a better induction argument, or a tighter concentration bound.
>
> 3. As we mentioned in the paper, the proof of Corollary 1 can be extended to arbitrary input distributions on the hypercube, thanks to the fact both Theorem 1 and 2 hold uniformly on pairs $x,y$ on the hypercube. As can be seen in lines 1211-1212, the expectation over the input distribution doesn't have an impact on the bound, since we only use the fact that integrates to one (thanks to the uniform bounds provided by the previous results).

---

### Official Review · Reviewer_jtsR · 2025-10-31

**Soundness:** 3
**Presentation:** 3
**Contribution:** 2
**Rating:** 4
**Confidence:** 4

**Summary:**

This paper focuses on noise sensitivity and quantifies the bound on the stability of SNNs. A notion of spectral simplicity is introduced, motivated by this bound. Several experiments are conducted to verify the proposed method.

**Strengths:**

1. The abstract is clear and well written.
2. The results shown in Figure 3 are interesting.

**Weaknesses:**

1. Experiments are very simplified and the scope is limited
2. The format of the paper can be further improved. Section 1.2 (Notion) in the Introduction section may not be appropriate, and it may be better to put it in Section 2. The bolded "Future directions" in the Conclusion section is confusing. Is it a section/subsection title? Why is a term bolded and put there? Suggesting to revise it to make the format coherent in the paper.
3. The conclusion does not summarize the results from numerical experiments. It will be beneficial to add them here to cover all key contents of the paper.

**Questions:**

1. Figure 3. onMNIST should be "on MNIST"
2. Could you further explain what the conclusion is for the expected noise sensitivity data shown in Figure 1? Does this Figure just show sIF and IF neuron has similar noise sensitivity? How are the contents related to the summary of Section 5 on line 375 to line 377?
3. "Notice that training reduces the model’s sensitivity,
but less strongly compared to MNIST. This aligns with the fact that both the training and test errors
are larger for CIFAR-10 (which achieves 84.38% training accuracy).
What are the training and test errors described here?

---

> ### Author Response · Authors · 2025-11-21
>
> We thank the reviewer for taking the time to evaluate our work, and their feedback. We answer his questions and comment on the weaknesses below.
>
> **Comment on weaknesses**
>
> 1. We perform additional experiments on neuromorphic MNIST and with the evaluation for randomly dropping 5 % of spikes, as suggested by Reviewer 77v2. This strengthen our experimental section. We added those experiments in the revised version.
>
> 2. The bolded text is a standard paragraph organization, used in several other works, to separate summary with future directions. The same holds for the Notation section as subsection of the Introduction. However, if the reviewer find it confusing, we are open to move the Notation at the beginning of Section 2, and to replace the Future directions paragraph with a dedicated subsection.
>
> 3. We add a brief summary of the experiments in the Conclusion, in this updated version. We highlight the text in blue color.
>
> **Answers to the questions**
>
> 1. Thanks for noticing this. We corrected it.
>
> 2. In Figure 1, we evaluate the noise sensitivity of single sIF (in Fig. 1a) and IF (in Fig. 1b), for different times $t=1,..,10$. There was a small typo in the legend (due to python numbering starting from $0$). For both models, the trend in the noise sensitivity is similar, for increasing $n$, in the regime $\nu=\frac{1}{\sqrt{n}}$. We add the sIF theoretical bound in both to suggest that Theorem 1 would hold for IF neurons as well. We add a line explaining clarifying that in the current revised version. For a theoretical explanation, see our answer to Question 2 of Reviewer rx3u.
>
> 3. Our objective in that sentence was to convey that on MNIST, the training seems to reduce the sensitivity of the model (details are in the caption of Figure 3). The same is true for CIFAR-10 to a lesser extent. The line quoted by the reviewer, aim to suggest that better train and test accuracy improve also the stability in those examples. We specified, in the current revised version, that with error we mean accuracy (or equivalently the misclassification error).

---

> ### Comment · Reviewer_jtsR · 2025-11-24
>
> Thanks to the authors for the rebuttal and detailed replies. I will increase my score to 6 to reflect the rebuttal.
>
> W1. Thank you for the new figure and the added text (Lines 460–467). The results clearly show different noise-tolerance behaviors on N-MNIST and MNIST. This comparison strengthens the contribution of the paper, and I personally appreciate this addition, especially the comparison (Line 467).
>
> W2.1. I still think the Notation section would be better placed outside the Introduction. In the current Introduction, the Notion section is the last subsection, and there are no subsequent subsections that explicitly use the definitions from the Notation section. It might be clearer to move the Notation closer to the sections where it is actually used (e.g., Section 2 or Section 3).
>
> W2.2. Thank you for the explanation. If the rest of the paper consistently follows the hierarchy of \section, \subsection, and bold text, it might be better to change this paragraph into a subsection to follow this hierarchy. In the current version, it is slightly confusing to use a bold sentence to divide the content of the Conclusion section into two parts. However, this is only a minor issue, so please feel free to use whichever format you prefer.
>
> W3, Q1–Q3. Thank you for the revisions.
>
> Q2. Thank you for the clarification regarding the figure.

---

### Official Review · Reviewer_rx3u · 2025-11-09

**Soundness:** 3
**Presentation:** 3
**Contribution:** 2
**Rating:** 4
**Confidence:** 4

**Summary:**

This paper analyzes the stability of randomly initialized spiking neural networks (SNNs) with sign integrate-and-fire (sIF) neurons, providing stability boundaries for both individual sIF neurons and multilayer sIF networks. Theoretical analysis proves the stability of random SNNs. This paper also performs spectral analysis, demonstrating that randomly initialized SNNs tend to exhibit simple spectral structures with low-frequency components.

**Strengths:**

1. This paper investigates the stability of SNNs from the perspective of Boolean function analysis, offering a novel viewpoint for SNN stability research.
2. This paper presents a detailed theoretical analysis and derivation, demonstrating a solid technical foundation.

**Weaknesses:**

1. This paper assumes that the leakage parameter (membrane time constant) of LIF neurons $\beta=1$, thereby reducing LIF neurons to IF neurons. Compared to simple IF neurons, LIF neurons are more commonly used and exhibit more complex neuronal dynamics. This paper does not provide further analysis to demonstrate how $\beta$ affects the stability bounds.
2. This paper employs sign leaky integrate-and-fire (sLIF) neurons. This neural model extends Boolean functions to the discrete-time domain. The sLIF neuron produces values of -1/1, which differs significantly from the typical LIF neuron that fires 0/1 spikes. This paper does not provide stability bounds for the typical LIF model, which holds greater significance for SNN stability research.

**Questions:**

1. Please provide the stability bounds for a general $\beta$.
2. Please provide the stability bounds for the typical LIF model.
3. This paper employs Xavier initialization with variance set to 1/n. However, this initialization method is not specifically designed for SNNs and is unsuitable for activation values with non-zero means. I wonder how alternative initialization methods, such as Kaiming initialization or initialization methods specifically designed for SNNs like [1], would affect the stability bounds.

[1] Ding, Jianhao, et al. "Assisting Training of Deep Spiking Neural Networks With Parameter Initialization." *IEEE Transactions on Neural Networks and Learning Systems* (2025).

---

> ### Author Response · Authors · 2025-11-21
>
> We thank the reviewer for their time and the useful feedback. We address each question individually.
>
> **Answer to Question 1, about the leaky parameter**
>
> We outline below how the proof extends to the general case $\beta \neq 1$.
> The overall strategy follows the proof of Theorem 1, but the dependence on $T$ becomes worse.
> We believe this degradation is an artifact of the proof rather than a true limitation of the model.
> In our main results $T$ is treated as a constant relative to $n$, and under this regime the conclusions remain unchanged.
> We also correct a minor typo in the earlier version, which does **not** affect the results. We corrected in blue color in the current revised version.
>
> 1. **Base case.**
> The case $t=1$ is unchanged and does not depend on $\beta$.
>
> 2. **Spike condition at time $T$.**
> For $t>1$, the neuron spikes at time $T$ for input $x=(x_t)_{t\in[T]}$ if an event of the form
>
> $$
> w^\top \sum_{t=1}^{T-1} \beta^{T-t} x_t \pm\frac{\theta}{2} \sum_{t=1}^{T-1} \beta^{T-t-1} (s_{t-1}+1)\ge 0
> $$
> holds, with an analogous condition for $y$. This changes the structure of the disagreement event $s_T(x)\neq s_T(y)$.
>
> 3. **Key partitioning step.**
> The analogue of eq.~(12) becomes
>
> $$
> \mathbb{P}[s_T(x)\neq s_T(y)]= \mathbb{P}\left[
> s_T(x)\neq s_T(y), \sum_{t=1}^{T-1}\beta^{T-t-1}s_t(x)\neq \sum_{t=1}^{T-1}\beta^{T-t-1}s_t(y)\right]
> $$
>
> $$
> \qquad+\mathbb{P}\left[s_T(x)\neq s_T(y),\sum_{t=1}^{T-1}\beta^{T-t-1}s_t(x)= \sum_{t=1}^{T-1}\beta^{T-t-1}s_t(y)\right].
> $$
> Unlike the case $\beta=1$, the term
> $\sum_{t=1}^{T-1}\beta^{T-t-1}s_t(x)$
> can take $2^{T-1}$ distinct values, which is the main source of the suboptimal dependence on $T$.
>
> 4. **Averaged inputs.**
> We replace $\bar{x}_T,\bar{y}_T$ with
>
> $$\bar{x}^\beta_T := \frac{1}{T}\sum_{t=1}^{T-1} \beta^{T-t} x_t$$
>
> and
>
> $$\bar{y}^\beta_T:=\frac{1}{T}\sum_{t=1}^{T-1} \beta^{T-t} y_t.$$
>
> Define
>
> $$
> X^\beta:= \sqrt{n}w^\top \frac{\bar{x}^\beta_T}{|\bar{x}^\beta_T|}
> $$
>
> and
>
> $$
> Y^\beta := \sqrt{n}w^\top \frac{\bar{y}^\beta_T}{|\bar{y}^\beta_T|}.
> $$
>
> These are standard Gaussians with correlation
>
> $$
> \rho \ge 1 - \frac{|\bar{x}^\beta_T - \bar{y}^\beta_T|^2}
> {|\bar{x}^\beta_T||\bar{y}^\beta_T|}.
> $$
>
> By triangle inequality,
>
> $$
> |\bar{x}^\beta_T - \bar{y}^\beta_T|^2\le\left( \sum_{t=1}^{T} \frac{\beta^{T-t}}{T}\right)^2\max_{t\in[T]} h_t.
> $$
>
> 5. **Conditioning on spike-history patterns.**
> For each pattern $a=(a_t)_{t\in[T-1]}\in\lbrace 0,1\rbrace^{T-1}$, set
>
> $$
> \bar{a}^\beta_T:= \frac{1}{T}\sum_{t=1}^{T-1}\beta^{T-t-1}a_t.
> $$
>
> We must bound events of the form
>
> $$
> \mathcal{E}:=\left\lbrace
> X^{\beta} \ge \sqrt{n}\theta \frac{\bar{a}^{\beta}_T}{|\bar{x}^\beta_T|},\;
> Y^{\beta} <  \sqrt{n}\theta \frac{\bar{a}^{\beta}_T}{|\bar{y}^\beta_T|}
> \right\rbrace,
> $$
> plus the symmetric case.
>
> 6. **Two cases, as in the original proof.**
> We use the same case distinction from Theorem 1, since the conditions for Lemma 5 still hold for
> $\bar{x}^\beta_T$ and $\bar{y}^\beta_T$.
>
> - If $|\bar{x}^\beta_T|=|\bar{y}^\beta_T|=\mathcal{O}(\sqrt{n}/\log n)$, then
>
> $$
> \mathbb{P}[\mathcal{E}]\le e^{-\theta^2 \bar{a}^\beta_T \log^2 n}.
> $$
>
> - If $|\bar{x}^\beta_T|=|\bar{y}^\beta_T|=\omega(\sqrt{n}/\log n)$, the same argument works after replacing $t/T$ with $\bar{a}^\beta_T$, and $\bar{h}_t$ with
>
> $$(\sum^T_{t=1} \tfrac{\beta^{T-t}}{T})^2 \max_t h_t$$
>
> We also correct a typo in eq. (15), which should be an inequality, likewise for the first line of eq. (17). This does not affect the result.
>
> Summing over all $2^{T-1}$ patterns gives
> $$
> \mathbb{P}[s_T(x)\neq s_T(y)]
> \le
> C  2^{T-1}
> \theta
> \sqrt{\frac{\max_{t\in[T]} h_t \log^2 n}{n}}.
> $$
>
> 7. **Conclusion.**
> Since our main results assume $T$ is constant relative to $n$, the conclusions remain the same.
> Improving the dependence on $T$ would require a sharper analogue of eq. (15), which appears challenging due to the temporal correlations induced by the shared weights.
>
> We plan to include this as a formal remark with all the details after the discussion phase.

---

> > ### Author Response · Authors · 2025-11-21
> >
> > **Answer to the Question 2, on the typical LIF/Heaviside activations**
> >
> > We appreciate the reviewer’s question about extending our result to the Heaviside activation. The extension requires tracking the norms of the input vectors carefully, but the core argument is unchanged. For clarity, we outline the argument for the simpler case $t=1$; for $t>1$, the same reasoning applies exactly as in the proof of Theorem 1.
> >
> > 1. **Setup**:  Let $x,y\in\lbrace 0,1\rbrace^n$ with Hamming distance $h=d_H(x,y)=\mathcal{O}(\sqrt{n})$. Define
> >     $$
> >     X = w^\top x, Y = w^\top y,
> >     $$
> >
> >     and consider the disagreement event
> >     $$
> >     \mathcal{E} := \lbrace X\ge \theta\, Y<\theta\rbrace\cup\ \lbrace X<\theta\, Y\ge \theta\rbrace.
> >     $$
> >
> > 2. **Correlation structure** Partition the coordinates as
> >
> >     $$
> >     I=\lbrace i:x_i=y_i=1\rbrace,\quad
> >     J_1=\lbrace i:x_i=1,y_i=0\rbrace,\quad
> >     J_2=\lbrace i:x_i=0,y_i=1\rbrace,\quad
> >     J_3=\lbrace i:x_i=y_i=0\rbrace.
> >     $$
> >
> >     Then $h = |J_1|+|J_2|$, and
> >
> >     $$
> >     X \sim \mathcal{N}\left(0,\tfrac{|I|+|J_1|}{n}\right), Y \sim \mathcal{N}\left(0,\tfrac{|I|+|J_2|}{n}\right).
> >     $$
> >
> >     The normalized variables are standard Gaussians with correlation
> >
> >     $$
> >     \rho = \frac{|I|}{\sqrt{|I|+|J_1|}\sqrt{|I|+|J_2|}}.
> >     $$
> >
> >     Hence,
> >
> >     $$
> >     p := \mathbb{P}\left[\frac{\sqrt n}{\sqrt{|I|+|J_1|}}X\ge \theta\frac{\sqrt n}{\sqrt{|I|+|J_1|}}\;
> >     \frac{\sqrt n}{\sqrt{|I|+|J_2|}}Y < \theta\frac{\sqrt n}{\sqrt{|I|+|J_2|}}\right].
> >     $$
> > 3. **Case 1**: $|I|=o(n)$. Since $|J_1|+|J_2| = h = \mathcal{O}(\sqrt n)$, we have $|J_1|,|J_2| = \mathcal{O}(\sqrt n)$.
> >     Thus,
> >
> >     $$
> >     p \le \mathbb{P}[X \ge \theta]
> >       \le \exp\left(-\frac{\theta^2}{2}\frac{n}{|I|+|J_1|}\right),
> >     $$
> >
> >     which decays exponentially, mirroring Case 1 in the proof of Theorem 1.
> > 4. **Case 2**: $|I|=\omega(n)$. In this regime, $\rho$ is close to $1$. Applying Lemma~3 yields
> >
> >    $$
> >     p \le \sqrt{1-\rho^2}
> >         + \theta \frac{\sqrt n}{|I|+|J_1|}\frac{1-\rho}{\rho}
> >         + \theta \frac{\sqrt n||J_1|-|J_2||}{\sqrt{|I|+|J_1|}\sqrt{|I|+|J_2|}\rho}.
> >     $$
> >
> >     Using $|I|=\omega(n)$ and $h=\mathcal{O}(\sqrt n)$ gives
> >
> >     $$
> >     p = \mathcal{O}\left(\sqrt{\frac{h}{n}}\right),
> >     $$
> >
> >     which matches the scaling from our original proof. Hence the same stability bound holds for Heaviside activations.
> >
> > We plan to include this as a formal remark with all the details after the discussion phase.

---

> > > ### Author Response · Authors · 2025-11-21
> > >
> > > **Answer to Question 3, about different initializations**
> > >
> > > We agree with the reviewer that initialization in SNNs this need not be follows the same strategy as in ANNs and different scalings may be appropriate. This is an active topic of research in SNNs. Our proof adapts to those cases, but it might require some additional observations if one wants to optimize the bounds. Several popular intializations, including the Gaussian Kaimming case, consider a rescaled Gaussian variable. In that case, notice that when we apply the normalization to the random variables $X,Y$, we would divide by the variance, so an additional factor will appear. Optimizing the bound with respect to this variance would require adding variance dependent factors in the two case distinctions of Theorem 1 (Case 1 and Case 2). This is feasible but leads only to constant-level changes in the final bounds. The initialization proposed in the paper cited by the reviewer appears to fall precisely into this category (as expressed after eq.(21) in that reference), corresponding to a rescaled Gaussian variance. We also note that non-centered Gaussians are admissible to some extent (see our response to Reviewer 77v2).
> > > If the initialization distribution is non-Gaussian (e.g., uniform), then the analysis requires either (i) a replacement for Lemma 3 under that distribution, or (ii) invoking a CLT approximation for large $n$, since the relevant quantity is a weighted sum of i.i.d. variables. In both cases, the same qualitative conclusion should still hold.

---

> > > > ### Comment · Reviewer_rx3u · 2025-11-22
> > > >
> > > > Thanks for your detailed reply. My concerns have been addressed. I am willing to raise my rating.

---

### Official Review · Reviewer_77v2 · 2025-11-11

**Soundness:** 3
**Presentation:** 3
**Contribution:** 3
**Rating:** 6
**Confidence:** 4

**Summary:**

This paper investigates the theoretical stability properties of discrete-time Leaky Integrate-and-Fire (LIF) Spiking Neural Networks (SNNs). The authors employ Boolean function analysis to quantify noise sensitivity and connect it to a new notion of spectral simplicity, showing that wide, randomly initialized SNN classifiers are on average stable and biased toward low-frequency (simple) functions. They provide analytical bounds for both single-neuron and multi-layer cases (Theorems 1–2), establish a formal relationship between stability and Fourier spectrum concentration (Corollary 1), and validate the theoretical insights through small-scale simulations on MNIST and CIFAR-10.

**Strengths:**

Applying Boolean function tools to spiking dynamics is original and mathematically elegant. It connects neuromorphic computation with the mature literature on noise sensitivity and Fourier analysis. The paper is clear, with rigorous definitions (noise sensitivity, spectral concentration) and clean proofs for Theorems 1–2

**Weaknesses:**

1. The paper shows minimal experiments and primarily confirm qualitative trends. The paper would benefit from convincing demonstration of the claimed real world persistence of spectral simplicity in trained SNNs.

2. The experiments measure noise sensitivity directly but there are no empirical validation of spectral concentration using Fourier spectrum.
to verify spectral concentration empirically.

3. Can we also claim that spectral stability will imply parameter stability? For example, may be authors could try injecting Gaussian noise into weights or thresholds and measure output sensitivity with the model.

4. It would be nice to add evaluation for the output with randomly dropping 5 % of spikes or shortening the readout horizon TTT. C

**Questions:**

1. Could the author add any real world event benchmark like DVS-CIFAR10, SHD or larger networks to strengthen the empirical section.

2. Could authors add an example showing the power spectrum to substantiate the “spectrally simple” claim?

3. Would be nice to compute for tiny inputs, the cumulative spectral energy vs degree to visually confirm low-frequency dominance.

---

> ### Author Response · Authors · 2025-11-21
>
> We thank the reviewer for their time and the useful feedback. We address each weakness and question individually.
>
> **Comments on Weaknesses**
>
> 1. The objective of this paper is mainly theoretical: the contribution is to introduce the notions of expected noise sensitivity (ENS) and spectral simplicity for SNNs and to provide analytical bounds on the ENS, which directly lead to Fourier spectrum concentration. The experimental section is intended solely to support the presented results and to observe how the training dynamics affect the ENS, rather than serving as a comprehensive empirical study of SNN stability. We agree with the reviewer that a further investigation of the spectral simplicity of SNNs across different architectures, training schemes, and realistic tasks is an important direction, but it would constitute a substantial study of its own (see answer to question 2), and it falls outside the scope of this paper.
>
> 2. We comment on this point in the answer to Question 2 below.
> 3. Although parameter stability is not the main focus of our paper, several elements of our analysis can be adapted to that setting, with some caveats. However, we want to stress that the results do not carry over verbatim, and a reformulation is required. Below, we outline how the first steps of our proof strategy (as in Theorem 1) extend to how we interpret the weight perturbations suggested by the reviewer. Let $w_\star \in \mathbb{R}^n$ be the parameters of a LIF-SNN neuron as in Definition~1. Consider a perturbed neuron with weights $w \sim \mathcal{N}(w_\star, \frac1{n})$. For fixed inputs $x,y \in \{-1,1\}^n$, the probability of disagreement is
> $$\mathbb{P}[\operatorname{sign}((w+w_\star)^\top x - \theta)\neq \operatorname{sign}((w+w_\star)^\top y - \theta)].$$
> Let $X = w^\top x$ and $Y = w^\top y$. These are centered standard Gaussians with correlation $\rho = \langle x/\|x\|,\, y/\|y\|\rangle$. The disagreement event is $\mathcal{D} := \lbrace X > \theta - w_\star^\top x,\; Y \le \theta - w_\star^\top y\rbrace \cup\lbrace X \le \theta - w_\star^\top x,\; Y > \theta - w_\star^\top y\rbrace$. Our Lemma 3 applies directly with $a =\theta- w_\star^\top x$, $b =\theta- w_\star^\top y$, yielding
> $$\mathbb{P}[\mathcal{D}]\le 2\left(\sqrt{1 - \rho^2}+ |w_\star^\top x|\,\frac{1-\rho}{\rho}+ \frac{|w_\star^\top(x-y)|}{\rho}\right).$$
> In the regime $|x-y|^2_2 = \mathcal{O}(\sqrt{n})$ used in our main results, one has $\rho = 1 - \mathcal{O}(n^{-1/2})$. Then $\mathbb{P}[\mathcal{D}] \to 0$ as $n \to \infty$ if $|\theta-w_\star^\top x|(1-\rho)\to 0$ and $|w_\star^\top(x-y)| \to 0$. For the second condition, by Cauchy-Schwarz, $|w_\star^\top(x-y)| \le|w_\star| |x-y| =\mathcal{O}(|w_\star| n^{1/4})$,
> so we require $|w_\star| = o(n^{-1/4})$. Similarly, by Cauchy-Schwarz, triangle inequality and $1-\rho=\mathcal{O}(\frac1{\sqrt{n}})$, we get
> $$|\theta-w_\star^\top x|(1-\rho)\leq \theta(1-\rho)+|w_\star| \sqrt{n}(1-\rho)=\mathcal{O}(\theta/{\sqrt n})+\mathcal{O}(|w_\star|).$$
> Hence, if $|w_\star|=o(1)$, then $|\theta-w_\star^\top x|\,\frac{1-\rho}{\rho}\to 0$. Combining both, it is sufficient to have convergence to $0$, that $|w_\star| = o(n^{-1/4})$ holds. While it is not clear that this is optimal, a condition of this type is natural: if $|w_\star|$ were much larger, the neuron would spike almost always (or almost never) on a substantial fraction of inputs, and the stability would be trivial. In our random-weight setting $w \sim \mathcal{N}(0, 1/n)$, the variance scaling is chosen to avoid these degenerate cases. A similar extension is possible for random thresholds, by conditioning on the threshold and controlling an additional integral, which would require a generalization of Lemma 3. We view this as an interesting direction for future work.
>
> 4. In Table 1,  we show the noise sensitivity of a random sLIF and LIF spiking neural network with respect to 5 \% random dropping of the input signal, meaning that we set to zero 5\% of the input signal components chosen randomly. The result shows dropout leads to smaller output sensitivity than random flipping. We invite the reviewer to check the revised version, changes are highlighted in blue color.

---

> > ### Author Response · Authors · 2025-11-21
> >
> > **Answer to Questions**
> >
> > 1. To complement the experimental section as suggested by the reviewer, we add in this revised version experiments in the NMNIST dataset. We invite the reviewer to check the revised version, changes are highlighted in blue color.
> >
> > 2. We agree with the reviewer that providing empirical evidence of spectral concentration would strengthen the paper. We would like to emphasize, however, that estimating Fourier--Walsh coefficients in high dimensions is intrinsically challenging. For a classifier $f : \lbrace-1,1\rbrace^n \to \lbrace-1,1\rbrace$, the coefficients are defined as
> >  $$\hat{f}(S) = \mathbb{E}_{x \sim \mathrm{Unif}(\{-1,1\}^n)}[ f(x)\chi_S(x)]$$
> >
> >     and $ \chi_S(x)=\prod_{i\in S} x_i$.
> >     Accurately approximating these expectations via Monte Carlo requires an exponentially large number of samples, analogous to high-dimensional numerical integration. More refined methods, such as the Kushilevitz-Mansour (KM) algorithm, allow identification of *large* coefficients, but they do not scale favorably with the ambient dimension. We are currently trying to obtain reliable estimates of the coefficients, but obtaining good approximations seems to be a research problem in itself. We will appreciate any insights, references, or suggestions that we may have overlooked.
> >
> >
> > 3. We are not completely sure what the reviewer intended with this question. The Fourier-Walsh spectrum is a property of the function $f$ itself and does not depend on specific input points. If the reviewer is referring to low-dimensional projections or to the spectrum of restricted versions of $f$, we would be grateful for clarification so we can address the concern precisely.

---

### Author Response · Authors · 2025-12-02
**Summary of the rebuttal process**

We thank again all the reviewers for their engagement and their insightful discussion. We are happy for the overall positive evaluation of our work. We are glad that we could address most of the reviewers’ questions. Unfortunately, we could not finish the discussion with Reviewer KsVi, but we believe that our follow-up answer posted below clarifies the remaining open point.

Below, we summarize the overall rebuttal.

1. We provided additional experiments on neuromorphic datasets, further supporting our theoretical results (see Q1 of Reviewer 77v2).
2. We considered the suggestion of Reviewer jtsR regarding the \subsection vs. \paragraph organization.
3. We clarified technical points of the paper with the reviewers, in particular: (1) the extension of the results to Heaviside activations (see our answer to Q2 of Reviewer rx3u), (2) the general $\beta$ case (see answer to Q1 of Reviewer rx3u). We will add both points as formal remarks for the camera ready version, since the Reviewer rx3u was satisfied with the answers.
4. We discussed the technical challenges to get tighter bounds for deep models. (see our answer to Reviewer 7NY3)

---

### Meta-Review · Area_Chair_Ut59 · 2026-01-06

**Summary:**

This paper studies the stability/robustness of discrete-time LIF-style spiking neural network (SNN) classifiers through the lens of Boolean function analysis. The main theoretical message is that wide random LIF-SNNs are stable on average, and that this stability can be explained by concentration of the Fourier–Walsh spectrum on low-degree components. The paper further introduces a notion of spectral simplicity to formalize “simplicity” via spectral concentration and connects it to simplicity bias. Experiments (single neurons and multi-layer networks) are provided to support the theoretical claims and to examine how training affects noise sensitivity.

**Reviewer Concerns:**

Overall, the reviews converged positively after discussion. The rebuttal effectively addressed the main technical concerns:
- Model assumptions / relevance to standard LIF (Heaviside) and general leak parameter: Reviewers questioned the initial focus on signed/IF-like variants and simplified settings. The authors provided detailed arguments for extending the analysis to (i) general leak parameter regimes and (ii) typical Heaviside/LIF activations, clarifying which proof steps change and why the qualitative conclusions remain. This resolved the key “theory–practice gap” concern for multiple reviewers, and at least one reviewer explicitly indicated willingness to raise their rating after these clarifications.
- Empirical strengthening: The authors added additional experiments on a neuromorphic dataset (e.g., N-MNIST) and provided an extra stability test (e.g., input dropout), improving the empirical support relative to the original submission and aligning with reviewer requests.
- Fourier spectrum empirical estimation: Reviewers requested empirical Fourier probes to validate spectral concentration directly. The authors explained why estimating Fourier–Walsh coefficients in high dimensions is intrinsically challenging and why naive Monte Carlo is infeasible, which is a reasonable limitation for the scope of this theoretical work.
- Scope with structured connectivity: A follow-up question raised whether structured/recurrent connectivity invalidates the approach. The authors gave a concrete explanation of where independence/Gaussianity assumptions break under correlated weights and clarified that extensions would require different techniques. This is a helpful and honest limitation clarification.

The work is primarily theoretical and focuses on random networks; applicability to structured/trained connectivity is not covered beyond discussion, and extending the proof techniques to correlated weights remains open.
Empirical validation is still limited in scale and does not directly measure the Fourier spectrum concentration in high dimension (though the authors provided a reasonable explanation for the difficulty and added requested experiments).
Some presentation/organization issues were noted (notation placement, formatting), but these are minor and can be addressed in the camera-ready.

**Reviewer Scores:**

Reviewer 7NY3: Already strongly positive at submission time (score 8). The reviewer raised mainly forward-looking questions about tightening bounds and extensions. The rebuttal provided clarifications and limitations without revealing new issues. The score would remain a strong accept (8).

Reviewer 77v2: Initially marginally above the acceptance threshold (score 6). The reviewer’s main concerns were about the limited empirical section and lack of direct spectral measurements. The authors clarified the theoretical scope of the paper and added neuromorphic dataset experiments, which reasonably addressed these concerns. The score would remain at a weak accept / borderline accept (6).

Reviewer rx3u: Initially marginally below threshold (score 4), with technical concerns regarding neuron models (sLIF vs. standard LIF), leak parameter assumptions, and initialization. These points were addressed in detail during the discussion, and the reviewer explicitly indicated that their concerns were resolved and that they were willing to raise their rating. The score would increase to a weak accept (6).

Reviewer jtsR: Initially marginally below threshold (score 4), mainly due to limited experiments and presentation issues. After the rebuttal, the reviewer explicitly increased their score, noting that the added experiments and clarifications strengthened the contribution. The score would increase to a weak accept (6).

Reviewer KsVi: Initially marginally above threshold (score 6), with questions about scope, clarity of contributions versus prior work, and applicability to structured connectivity. The discussion clarified these points and explicitly identified limitations. While some extensions remain open, the reviewer’s technical questions were answered. The score would remain a weak accept (6).

---

### Decision · Program_Chairs · 2026-01-26

Accept (Poster)